## REPORT

# Prolonged depletion of profilin 1 or F-actin causes an adaptive response in microtubules

Bruno A. Cisterna[1], Kristen Skruber[2], Makenzie L. Jane[1], Caleb I. Camesi[1], Ivan D. Nguyen[1], Tatiana M. Liu[1], Peyton V. Warp[3], Joseph B. Black[4], Mitchell T. Butler[5,6], James E. Bear[5,6], Danielle E. Mor[1], Tracy-Ann Read[1], and Eric A. Vitriol[1]

In addition to its well-established role in actin assembly, profilin 1 (PFN1) has been shown to bind to tubulin and alter microtubule growth. However, whether PFN1's predominant control over microtubules in cells occurs through direct regulation of tubulin or indirectly through the polymerization of actin has yet to be determined. Here, we manipulated PFN1 expression, actin filament assembly, and actomyosin contractility and showed that reducing any of these parameters for extended periods of time caused an adaptive response in the microtubule cytoskeleton, with the effect being significantly more pronounced in neuronal processes. All the observed changes to microtubules were reversible if actomyosin was restored, arguing that PFN1's regulation of microtubules occurs principally through actin. Moreover, the cytoskeletal modifications resulting from PFN1 depletion in neuronal processes affected microtubule-based transport and mimicked phenotypes that are linked to neurodegenerative disease. This demonstrates how defects in actin can cause compensatory responses in other cytoskeleton components, which in turn significantly alter cellular function.

## Introduction

Actin is a protein that forms polarized, linear polymers that are the principal force-generating agents within a cell. One of the main functions of the actin cytoskeleton is to control cell shape by modulating cortical tension in response to vital processes such as mitosis and migration (Chugh and Paluch, 2018; Murrell et al., 2015). Actin can also produce protrusive forces by polymerizing multiple actin filaments or contractile forces through engagement with molecular motors such as non-muscle myosin II (Vicente-Manzanares et al., 2009; Walker et al., 2020). However, actin is just one component of a cell's cytoskeleton, which also includes microtubules and intermediate filaments. Each of these polymer networks has their own mechanical properties, assembly/disassembly dynamics, and regulatory molecules (Pollard and Goldman, 2018). Executing normal cellular functions requires coordination of the cytoskeletal elements through various crosstalk mechanisms (Seetharaman and Etienne-Manneville, 2020).

Actin crosstalk may occur through crosslinking proteins that bind multiple types of filaments, such as plectin (Fontao et al., 2001; Na et al., 2009), tau (Elie et al., 2015; Ramirez-Rios et al., 2016), spectraplakins (Brown, 2008; Voelzmann et al., 2017), or drebrin/Eb3 pathway (Geraldo et al., 2008; Poobalasingam et al.,

2022). Some of these coupling proteins can also act as polymerases and link the growth of different networks to each other (Henty-Ridilla et al., 2016; Preciado López et al., 2014). Cytoskeletal crosstalk can also happen through signaling molecules that bind one type of cytoskeletal element yet regulate the dynamics of another, such as the microtubule-binding RhoA activator GEF-H1, which induces actomyosin contractility when microtubules disassemble (Krendel et al., 2002). Finally, cytoskeletal crosstalk can also occur independently of a direct molecular connection, for example, when one cytoskeletal network serves as a barrier or a channel to the growth or localization of another (Schaefer et al., 2002). Because both direct and indirect forms of crosstalk may happen simultaneously, it can be difficult to determine which is predominant without precise experimental approaches that differentiate between the different mechanisms.

Recently, profilin 1 (PFN1), an actin monomer binding protein that is crucial for filament assembly (Carlsson et al., 1977; Pernier et al., 2016; Skruber et al., 2018, 2020), was shown to interact with tubulin and alter microtubules polymerization (Henty-Ridilla et al., 2017; Nejedla et al., 2016). In solution, PFN1 binds both the soluble tubulin dimer and the microtubule lattice

[1]Department of Neuroscience and Regenerative Medicine, Medical College of Georgia at Augusta University, Augusta, GA, USA; [2]Department of Cellular and Molecular Pharmacology, University of California San Francisco, San Francisco, CA, USA; [3]University of Miami Miller School of Medicine, Miami, FL, USA; [4]Division of Urologic Surgery, Beth Israel Deaconess Medical Center, Boston, MA, USA; [5]Department of Cell Biology and Physiology, University of North Carolina at Chapel Hill School of Medicine, Chapel Hill, NC, USA; [6]Lineberger Comprehensive Cancer Center, University of North Carolina at Chapel Hill School of Medicine, Chapel Hill, NC, USA.

Correspondence to Tracy-Ann Read: tread@augusta.edu; Eric A. Vitriol: evitriol@augusta.edu.

and accelerates the polymerization rate (Pimm et al., 2022). In cells, PFN1 has also been coimmunoprecipitated with tubulin and the gamma-tubulin complex (Nejedlá et al., 2020; Nejedla et al., 2016), and there are microtubule phenotypes associated with PFN1 depletion (Bender et al., 2014) and mutation (Henty-Ridilla et al., 2017; Pinto-Costa et al., 2020). However, the mechanisms of how PFN1 affects microtubules in cells remain unclear, and there is conflicting evidence of whether it enhances (Henty-Ridilla et al., 2017; Pimm et al., 2022; Pinto-Costa et al., 2020) or inhibits (Bender et al., 2014; Nejedlá et al., 2020; Nejedla et al., 2016) microtubule growth. Given PFN1's crucial role in actin assembly, it is highly likely that at least some of its effects on microtubules occur indirectly through actin. Here, we show that PFN1-dependent actin assembly controls microtubule stabilization through actomyosin contractility. These effects are enhanced in neuronal processes, where depletion of PFN1, F-actin, or myosin activity causes cytoskeletal changes resembling those found in neurodegenerative disease.

## Results and discussion

### PFN1 KO causes an adaptive response in microtubules that can be reproduced by prolonged depletion of F-actin

We investigated the relationship between actin and microtubules in cath.A.differentiated (CAD) cells lacking PFN1 (PFN1 KO) by immunostaining for α-tubulin and actin filaments (F-actin). Previously, we established that removing PFN1 expression in CAD cells results in ~50% reduction of total F-actin (Skruber et al., 2020). Here, we saw that PFN1 KO cells had significantly more microtubules than controls in CAD cells (Fig. 1, A and B) and MEFs cells (Fig. S1), which matches previous reports of PFN1-deficient cells (Bender et al., 2014; Nejedlá et al., 2020; Nejedla et al., 2016). Interestingly, there was a significant inverse correlation between microtubule and F-actin levels in PFN1 KO CAD cells but not in controls (Fig. 1 C). This suggested a crosstalk mechanism occurring after a threshold value of F-actin loss had been surpassed.

To further explore whether the increase in microtubules in PFN1-deficient cells was caused by loss of direct regulation by PFN1 or indirectly through F-actin depletion, we performed experiments in PFN1 KO CAD cells rescued with GFP labeled wild-type (WT) and mutant PFN1, including the actin-binding deficient R88E, the ALS-causative PFN1 variants M114T and G118V, and the ALS risk factor E117G (Sohn et al., 1995; Wu et al., 2012). Previous work showed that the ALS-associated mutants, but not R88E, perturb PFN1's ability to alter microtubule growth (Henty-Ridilla et al., 2017). While WT PFN1 completely restored F-actin and microtubule levels to those found in control cells, the PFN1 mutants did not (Fig. 1, D–F). Interestingly, the ability of the mutants to decrease the number of microtubules was directly proportional to their ability to rescue actin polymerization (Fig. 1 F). For example, E117G was able to mostly restore actin assembly and reduce microtubules, but expressing R88E, M114T, or G118V in PFN1 KO cells had essentially no effect on either parameter (Fig. 1, D–F). Similarly, only PFN1 KO cells expressing WT and E117G, whose levels of F-actin and tubulin were at least partiality restored, lost the correlation between the amount of F-actin and microtubules, supporting the assumption that this

correlation is evident only under a certain F-actin threshold (Fig. 1 G). Interestingly, the causative ALS-linked mutations have been previously shown to have both loss and gain of function effects on PFN1's ability to polymerize actin, depending on the specific assay that was used. These include reducing PFN1's affinity for actin monomers and enhancing its ability to assemble formin-mediated filaments (Liu et al., 2022; Schmidt et al., 2021). However, their effects on total cellular levels of F-actin in our assays were remarkably similar and demonstrated a near-complete loss of function.

Since both microtubule-binding proficient (R88E) and deficient (M114T and G118V) mutants failed to rescue PFN1 KO microtubule phenotypes, and restoration of microtubules was directly proportional to each PFN1 variant's ability to rescue actin assembly, we hypothesized that the predominant role of PFN1 in microtubule regulation in these cells was through actin assembly. To test this, we treated control cells with low doses of Latrunculin A (Lat A), a monomeric actin-binding drug that inhibits polymerization (Fujiwara et al., 2018), to mimic the loss of actin filaments seen in PFN1 KO CAD cells. We found that a treatment of 10 nM Lat A reduced actin polymerization by ~40% without significantly altering cell morphology, even when administered overnight (Fig. 2 A). Interestingly, applying Lat A to cells for 3 h did not affect microtubule assembly or organization. However, an overnight Lat A treatment reproduced the PFN1 KO microtubule phenotype (Fig. 2, B and C). That changes were only visible after long-term depletion of F-actin was suggestive of a homeostatic response of the microtubule cytoskeleton rather than a transient signaling mechanism induced by a microtubule-regulating F-actin binding protein.

PFN1 depletion has been shown to cause the acetylation of α-tubulin on lys40 (Nejedla et al., 2016). This posttranslational modification is associated with aged microtubules and is thought to make them more curved, flexible, and resistant to breakage (Eshun-Wilson et al., 2019; Janke and Montagnac, 2017). Furthermore, microtubules become hyperacetylated in response to increased and sustained mechanical forces (Portran et al., 2017; Xu et al., 2017), which they may be subjected to if the actin filaments that maintain cell shape and stiffness are significantly reduced (Janke, 2014; Li and Yang, 2015). We measured acetyl α-tubulin levels using both immunocytochemistry (Fig. 2, D and E) and immunoblotting (Fig. 2 F) and found that knocking out PFN1 or treating cells overnight with Lat A caused an identical increase in acetyl α-tubulin (Fig. 2, D–F). Importantly, we measured the amount and acetylation of microtubules in control PFN1 KO mouse embryonic fibroblasts (Fig. S1) and obtained nearly identical results to those from CAD cells, indicating that these changes are not cell-type specific.

In conclusion, while there may be subtle effects on microtubules from PFN1 directly binding to tubulin that are missed in our assays, our results (Fig. 1 and Fig. 2) provide strong evidence that PFN1 depletion predominantly alters microtubules through an adaptive response caused by the long-term loss of F-actin.

### Changes to microtubules in response to PFN1 depletion are enhanced in the processes of differentiated CAD cells

Neurites are long, thin cytoskeletal structures specialized for the long-range transport of material from the cell body to the

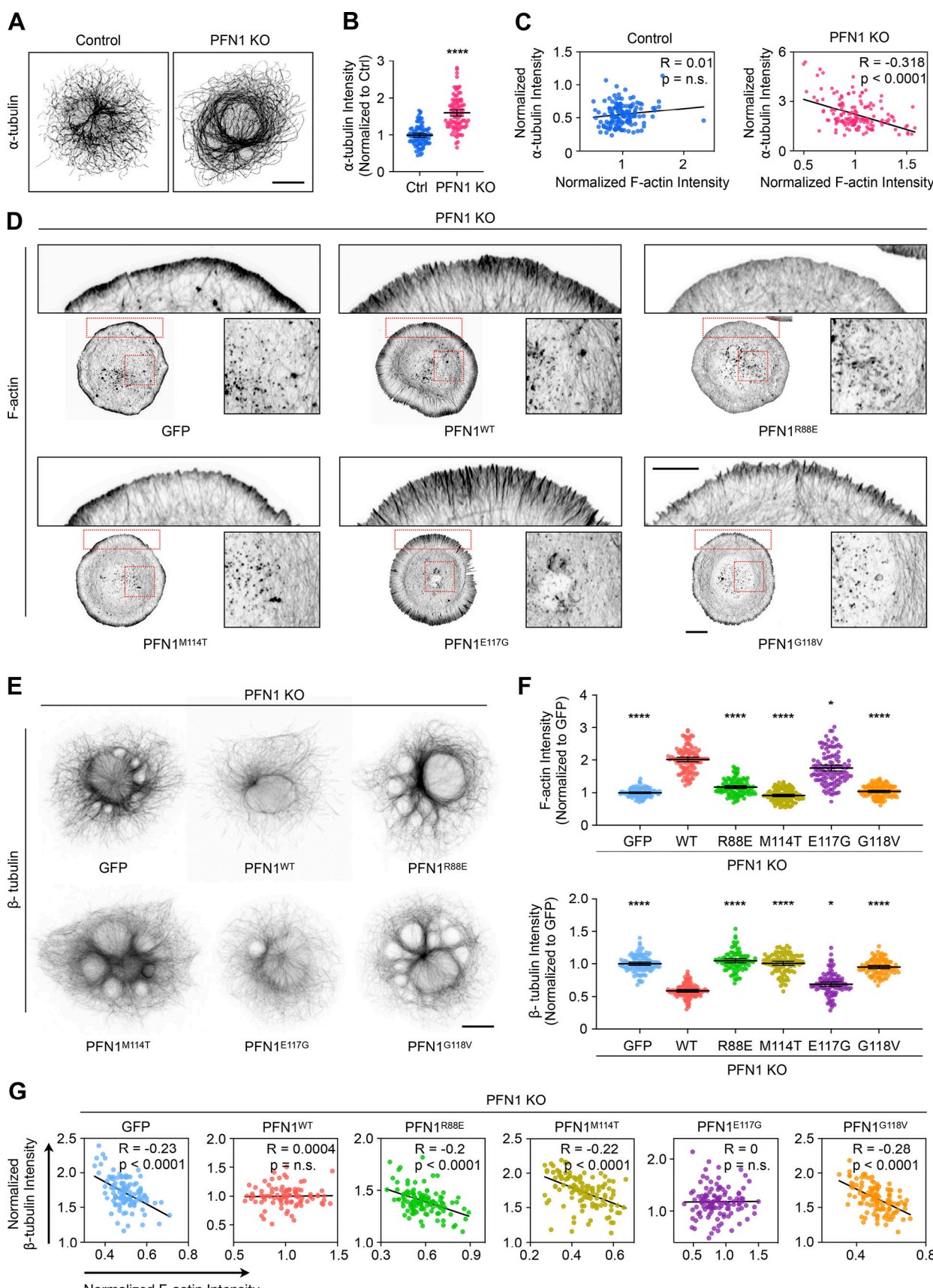

Figure 1. **PFN1 KO causes an adaptive response in the microtubule cytoskeleton. (A)** Representative images of α-tubulin in Control and PFN1 KO CAD cells. Scale bar: 10 μm. **(B)** Quantification of mean α-tubulin fluorescence intensities in A. Data are normalized to Control (Ctrl) and plotted as mean ± 95% CI.

$n$ = 97 cells for Control and $n$ = 96 cells for PFN1 KO. Significance was calculated using a two-sided Student's $t$ test. **(C)** Correlation between F-actin and α-tubulin intensities for cells in A. Intensities were normalized to the mean of each dataset. **(D and E)** Representative images of F-actin (D) and β-tubulin (E) in PFN1 KO CAD cells transfected with GFP, GFP-PFN1$^{WT}$ (PFN1$^{WT}$), GFP-PFN1$^{R88E}$ (PFN1$^{R88E}$), GFP-PFN1$^{M114T}$ (PFN1$^{M114T}$), GFP-PFN1$^{E117G}$ (PFN1$^{E117G}$), or GFP-PFN1$^{G118V}$ (PFN1$^{G118V}$). Insets in D highlight F-actin at the leading edge and cell body. Scale bar: 10 µm. Inset scale bar: 5 µm. **(F)** Quantification of mean F-actin and β-tubulin fluorescence intensities in E. Data were normalized to GFP and plotted as mean ± 95% CI. $n$ = 120 cells for each transfection. Significance was calculated against GFP using ANOVA and Dunnett's post hoc test. **(G)** Correlation between F-actin and β-tubulin intensities for cells in F. Intensities were normalized to the mean of each dataset. **** indicates P < 0.0001, * indicates P = 0.034, ns = not significant (P > 0.05).

synapse. We hypothesized that the actin-microtubule homeostasis we observed in undifferentiated CAD cells would be more relevant to neuronal processes since they are principal components of this structure (Coles and Bradke, 2015). To test this, we first induced differentiation in CAD cells, which conveniently differentiate into a neuronal-like morphology upon serum

withdrawal and form processes that are hundreds of microns long (Kapustina et al., 2016).

After 4 days of differentiation, the processes of PFN1 KO cells exhibited a similar reduction in F-actin levels (~50%) as observed in the undifferentiated CAD cells (Fig. 3, A and B). Furthermore, they were narrower than the control cells (Fig. 3, A

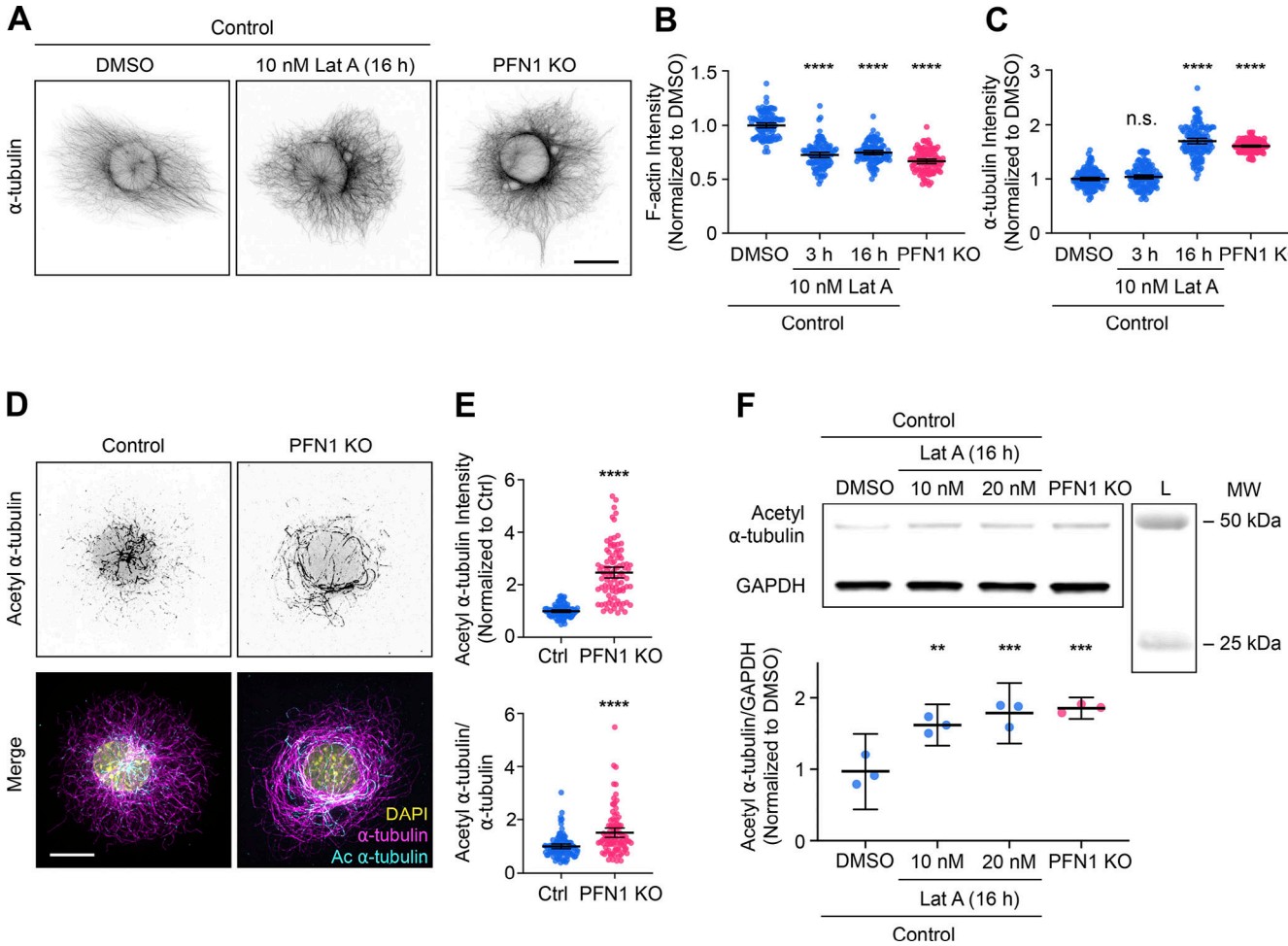

Figure 2. **Prolonged depletion of F-actin can reproduce the adaptive response in microtubules caused by knocking out PFN1. (A)** Representative images of α-tubulin in Control CAD cells incubated with 0 or 10 nM Latrunculin A (Lat A) for 16 h and PFN1 KO CAD cells. Scale bar: 10 µm. **(B and C)** Quantification of mean fluorescence intensities in A. F-actin (B) and α-tubulin (C). Data are normalized to Control and plotted as mean ± 95% CI. $n$ = 101 cells for each condition. Significance was calculated against control using ANOVA and Dunnett's post hoc test. **(D)** Representative images of acetyl α-tubulin at the top and merge images of DAPI (yellow), α-tubulin (magenta), and acetyl α-tubulin (cyan) at the bottom in control and PFN1 KO CAD cells. Scale bar: 10 µm. **(E)** Quantification of mean fluorescence intensities in D. Acetyl α-tubulin is at the top, and the acetyl α-tubulin/α-tubulin ratio is at the bottom. Data are normalized to control (Ctrl) and plotted as mean ± 95% CI. $n$ = 96 cells for Ctrl and PFN1 KO. Significance was calculated using a two-sided Student's $t$ test. **(F)** Western blot of acetyl α-tubulin and GAPDH in Control CAD cells incubated with 0, 10 nM, or 20 nM Lat A for 16 h, and PFN1 KO CAD cells are at the top, and quantification of levels expression at the bottom. Individual data normalized to control and plotted as mean ± 95% CI. $n$ = 3 independent experiments. Significance was calculated against control using ANOVA and Dunnett's post hoc test. **** indicates P < 0.0001, ** indicates P < 0.01. Source data are available for this figure: SourceData F2.

and B). Additionally, the filopodia were less dense and shorter in length than the control cells (Fig. 3 C). Despite similar decreases in actin polymerization, there was a significantly more substantial increase in both a-tubulin and acetylation of a-tubulin in the PFN1 KO processes (Fig. 3, D and E). In undifferentiated PFN1 KO CAD cells, the whole-cell measurement showed an increase of ~50% in microtubules (Fig. 1, A and B) and double the amount of acetyl α-tubulin (Fig. 2, D–F). However, measuring a region of interest in the processes of differentiated cells showed double the microtubules and tripled the amount of acetyl α-tubulin in PFN1 KO cells (Fig. 3, D and E). The PFN1 KO microtubule phenotype in processes was rescued by transfecting cells with WT, but not R88E, PFN1 (Fig. 3, F and G). Finally, whole-cell measurements of α-tubulin and acetyl α-tubulin by immunoblotting in differentiated CAD cells had a more moderate increase in PFN1 KO cells (Fig. 3, H and I), further indicating that the PFN1 depletion–induced change in microtubules and their acetylation is concentrated in their processes.

Maintaining neurites requires the antero- and retrograde transport of materials between the cell body and distal regions using microtubule motors. Since microtubule acetylation can alter the binding and motility of microtubule motors (Balabanian et al., 2017; Reed et al., 2006), we investigated whether changes to microtubules following long-term depletion of polymerized actin had functional consequences in the processes of differentiated PFN1 KO cells. We performed live cell imaging experiments to measure organelle transport. Despite having a similar distribution of mitochondria (Fig. 3 J), there were significant differences in mitochondria mobility in the processes of PFN1 deficient cells (Fig. 3, K and L), with PFN1 KO mitochondria exhibiting increased velocities (Fig. 3 L). Similar results were obtained in experiments measuring the mobility of lysosomes (Fig. 3, M and N). It appeared that increased mobility of organelles was found in both the anterograde and retrograde directions, which coincides with previous work showing that both kinesin and dynein may react similarly to hyperacetylated microtubule bundles (Alper et al., 2014; Balabanian et al., 2017; Reed et al., 2006). However, additional experiments will be needed to determine which motors are specifically affected. Interestingly, enhanced axonal transport has been linked to neurodegenerative diseases like ALS (Breuer et al., 1987; Hirano et al., 1984), Parkinson's (Prots et al., 2013; Utton et al., 2005), and Alzheimer's disease (Blagov et al., 2022; Wang et al., 2015). Here, we demonstrate that this also may be a consequence of PFN1 loss of function.

We also evaluated the intermediate filament neurofilament heavy chain, a well-known neuron marker and structural component of neuronal processes (Yuan and Nixon, 2021; Yuan et al., 2017). Previous RNAseq data revealed that CAD cells express all four neurofilament subunits (neurofilament light chain, neurofilament medium chain, neurofilament heavy chain, and peripherin) (Skruber et al., 2020). Immunocytochemistry experiments confirmed that neurofilament heavy chain was expressed in CAD but not in HeLa cells, a nonneuronal control (Fig. S2 A). In differentiated CAD cells, it is found throughout the entire process shaft (Fig. S2 B). Despite not

having a significant change in *NEFH* gene expression (Skruber et al., 2020), undifferentiated PFN1 KO cells showed an increase in neurofilament heavy chain protein (Fig. S2, A and C). The nascent processes of differentiated CAD cells had a similar increase in neurofilament heavy chain (Fig. S2 D). Interestingly, neurofilament accumulations are a hallmark of ALS (Itoh et al., 1992; Mizusawa et al., 1989; Poesen and Van Damme, 2019), and overexpression of this NEFH causes motor neuron disease in mice (Meier et al., 1999). These results highlight another potential pathogenic mechanism that PFN1 mutants could have in ALS.

## Prolonged depletion of F-actin increases the number and acetylation of microtubules in the processes of hippocampal neurons

We next wanted to determine if the same relationship between F-actin and microtubules existed in the processes of primary neurons. We cultured mouse E14 hippocampal neurons, treated them overnight with different concentrations of Lat A, and measured the relative amounts of F-actin, α-tubulin, and acetyl α-tubulin (Fig. 4, A and B). Mimicking the results we obtained in CAD cells, we found that F-actin decreases the α-tubulin and acetyl α-tubulin levels increase in neuronal processes (Fig. 4 B), with significant inverse correlations between local levels of F-actin and α-tubulin (Fig. 4 C), and F-actin and α-tubulin acetylation (Fig. 4 D). While it could be argued that the processes of PFN1 KO cells have so many microtubules because they began differentiation with an increased amount of them, these experiments demonstrate that microtubules can be substantially increased in mature neurites following overnight partial actin depolymerization. Furthermore, finding a near-identical phenotype in Lat A–treated hippocampal neurons addresses the concern that results obtained from CAD cells were caused by differential expression of microtubule regulators during differentiation.

## Inhibiting actomyosin contractility increases the number and acetylation of microtubules without depolymerizing F-actin in the processes of hippocampal neurons

Since effects on microtubules were only seen after overnight or permanent depletion of polymerized actin, we postulated that the microtubule cytoskeleton had adapted to long-term changes in the cells' mechanical properties (Li et al., 2023; Murrell et al., 2015). To determine if changes in cell mechanics could affect microtubules without depolymerizing actin, we treated cells overnight with the non-muscle myosin 2 inhibitor Blebbistatin, which reduces the cells' ability to maintain cortical tension and generate contractile actin structures (Kovács et al., 2004). Treating undifferentiated CAD cells with 15 μM Blebbistatin for 24 h had severe effects on cell morphology, which could be reversed after removing it and replacing the medium for 24 h (Fig. S3 A). As expected from previous work showing the effect of non-muscle myosin 2 loss of function on microtubule acetylation (Even-Ram et al., 2007; Joo and Yamada, 2014), Blebbistatin treatment caused a significant increase in the total α-tubulin acetylation measured by Western blot, which completely reverted to pretreatment levels following a 24-h wash-out (Fig. S3 B). However,

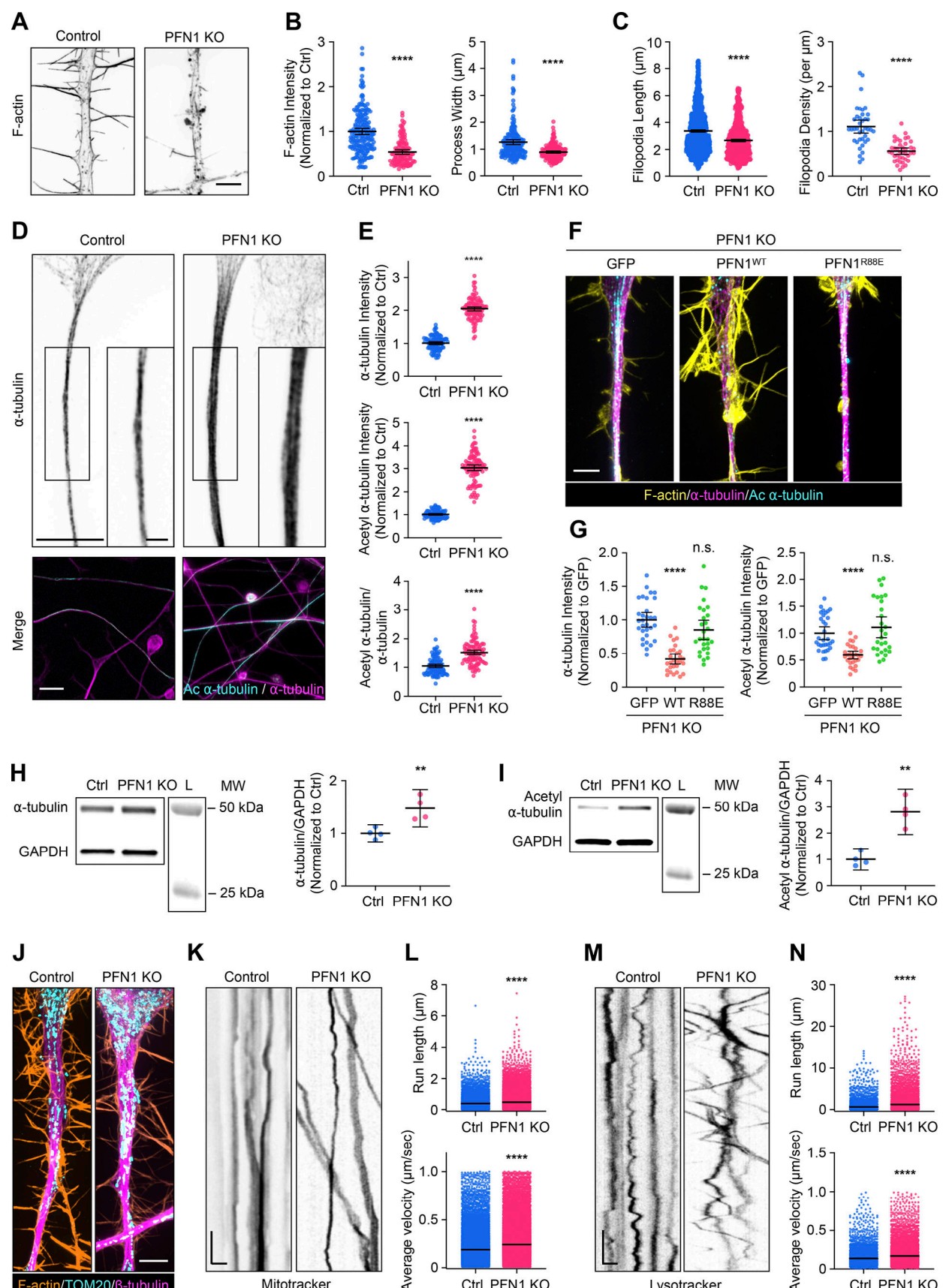

Figure 3. **The increase in acetylated microtubules caused by PFN1 KO is enhanced in the neuron-like processes of differentiated CAD cells and alters the active transport of organelles. (A)** Representative images of the F-actin in the processes of differentiated Control and PFN1 KO CAD cells. Scale bar:

4 µm. **(B)** Quantifications of mean F-actin fluorescence intensity and process width of A. F-actin intensities were normalized to control (Ctrl). Data are plotted as mean ± 95% CI. For F-actin intensity quantification, n = 200 processes for control and n = 120 for PFN1 KO. For process width quantification, n = 200 processes for control and n = 180 for PFN1 KO. Significance was calculated using a two-sided Student's t test. **(C)** Quantifications of filopodia length and density in A. For filopodia length quantification, n = 1,534 processes for control and n = 806 for PFN1 KO. For filopodia density quantification, n = 39 processes for control and n = 40 for PFN1 KO. Significance was calculated using a Mann–Whitney U test for filopodia length and a two-sided Student's t test for filopodia density. **(D)** Representative images of the α-tubulin at the top and merged images of α-tubulin (magenta) and acetyl α-tubulin (cyan) at the bottom in the processes of differentiated control and PNF1 KO CAD cells. Scale bar: 15 µm. Inset scale bar: 5 µm. **(E)** Mean fluorescence intensities in D. α-tTubulin intensity is at the top, acetyl α-tubulin intensity is in the middle, and the acetyl α-tubulin/α-tubulin ratio is at the bottom. Data are normalized to Ctrl (α-tubulin and acetyl α-tubulin) and plotted as mean ± 95% CI. N = 101 fields for Control and PNF1 KO cells. Significance was calculated using a two-sided Student's t test. **(F)** Representative merge images of F-actin (yellow), α-tubulin (magenta), and acetyl α-tubulin (cyan) in the processes of differentiated PFN1 KO cells transfected with GFP, GFP-PFN1^WT (PFN1^WT), and GFP-PFN1^R88E (PFN1^R88E). Scale bar: 4 µm. **(G)** Quantification of mean of α-tubulin and acetyl α-tubulin fluorescence intensities in F. Data were normalized to GFP and plotted as mean ± 95% CI. N = 31 processes for GFP, n = 28 processes for PFN1^WT and PFN1^R88E. Significance was calculated against GFP using ANOVA and Dunnett's post hoc test. **(H and I)** Western blot of α-tubulin (H) and acetyl α-tubulin (I) in Ctrl and PFN1 KO CAD cells at the top and quantification of levels expression at the bottom. Individual data were normalized to Ctrl and plotted as mean ± 95% CI. n = 4 independent experiments. Significance was calculated using a two-sided Student's t test. **(J)** Representative images of F-actin (orange), TOM20 (cyan), and β-tubulin (magenta) in processes of differentiated Control and PFN1 KO CAD cells. Scale bar: 4 µm. **(K and M)** Kymographs from mitochondria (Mitotracker) (K) and lysosome (Lysotracker) (M) in processes of differentiated Control and PFN1 KO CAD cells. Vertical scale bar: 5 s and horizontal scale bar: 5 µm. **(L and N)** Kymograph quantifications. Run length at the top and average velocity at the bottom for Mitotracker (L), and Lysotracker (N). Data are plotted as mean ± 95% CI. For Mitotracker run length quantification, n = 7,376 for Control and n = 26,016 for PFN1 KO. For Mitotracker average velocity quantification, n = 24,441 for Control and n = 42,989 for PFN1 KO. For Lyostracker run length quantification, n = 4,888 for Control and n = 7,534 for PFN1 KO. For Lysotracker average velocity quantification, n = 4,937 for Control and n = 7,612 for PFN1 KO. **** indicates P < 0.0001, ** indicates P < 0.01. Significance was calculated using Mann–Whitney U test. **** indicates P < 0.0001, ** indicates P < 0.01. Source data are available for this figure: SourceData F3.

Blebbistatin treatment did not affect tubulin levels measured using immunocytochemistry when actin was already low, either in PFN1 KO CAD cells (Fig. S3 C and D) or in control CAD cells under the effect of Lat A (Fig. S3, E and F). Conversely, activating actomyosin contractility with Calyculin A (Ishihara et al., 1989), which has been previously shown to decrease microtubule acetylation (Joo and Yamada, 2014), had no effect on the microtubules of PFN1 KO cells (Fig. S3, H and I). Together with the results detailed in Fig. 1 and Fig. 2, these experiments help to explain the connection between PFN1 loss of function and α-tubulin acetylation. When PFN1 is depleted, there is not sufficient F-actin for non-muscle myosin 2 to act upon and contractility is lost, instigating an adaptive response in microtubules that makes them less sensitive to damage in a mechanically compromised environment.

In the processes of hippocampal neurons, Blebbistatin treatment also increased the number of acetylated α-tubulin without altering F-actin levels (Fig. 5, A and B). As with undifferentiated cells, the effects of myosin II inhibition on microtubules were reversed after washing out Blebbistatin for 24 h. Also, inhibiting actomyosin contractility removed the inverse correlation between the amount or acetylation of microtubules and actin polymerization (Fig. 5 C). Like undifferentiated CAD cells, these results demonstrate how neuronal processes need myosin activity and the appropriate amounts of actin filaments to retain control over the amount, organization, and posttranslational modification of microtubules. This implies that defects in actin could cause pathogenic changes to microtubule-based processes in neurons, such as the dysregulated axonal transport that occurs in neurodegenerative disease (Blagov et al., 2022; Breuer et al., 1987; Hirano et al., 1984; Prots et al., 2013; Utton et al., 2005; Wang et al., 2015).

## Materials and methods
### Cell lines
Cath.-a-differentiated (CAD) cells: CAD cells (cat#CRL-11179; ATCC) were cultured in DMEM/F12 medium (cat#11330/032; Gibco) supplemented with 8% fetal bovine serum (FBS), 1% L-Glutamine, and 1% penicillin-streptomycin in standard tissue culture incubator. Profilin 1 knock-out (PFN1 KO) and Control (Ctrl) cells were generated from CAD cells with CRISPR/Cas9 (Skruber et al., 2020). CAD cells were differentiated under serum-free conditions for 4 days (Qi et al., 1997).

Mouse embryonic fibroblasts (MEFs): PFN1 KO MEF clonal lines were established from ARPC2 conditional knock-out mice (Rotty et al., 2015). Among these clonal lines, JR20s were used to express Cas9 and sgRNA (5′-TCGACAGCCTTATGGCGGAC-3′) targeting mouse PFN1 (Skruber et al., 2020) from pLenti-CRISPRv2 (#52961; Addgene) by lentiviral transduction. Lentivirus was generated by transfecting the plasmids pCMV-V-SVG (#8454; Addgene), pRSV-REV (#12253; Addgene), pMDLg/pRRE (#12251; Addgene), and pLentiCRISPRv2-PFN1sgRNA (500 ng each) into HEK293FT cells using X-tremeGENE HP DNA Transfection Reagent (Sigma-Aldrich). Lentivirus was harvested at 72 h to infect JR20 cells with 4 µg/ml of Polybrene. Around 72 h after the lentivirus infection, JR20 cells expressing Cas9 and PFN1 sgRNA were selected using 2 µg/ml puromycin for 48 h. LentiCRISPRv2-PFN1 (PFN1 LV) and control cells were cultured in Dulbecco's modified Eagle's medium (DMEM, cat#12430-047; Gibco) supplemented with 10% FBS and 1% penicillin-streptomycin in a standard tissue culture incubator. Cells were routinely tested for mycoplasma using the LookOut Mycoplasma PCR Detection Kit (cat#MP0035; Sigma-Aldrich).

### Primary hippocampal neuron cultures
Mouse primary hippocampal neuronal cultures were prepared as previously described (Encalada et al., 2011). Briefly, hippocampi were dissected from P2 mouse pups in cold Hanks' balanced salt solution (HBSS) supplemented with 0.08% D-glucose (Sigma-Aldrich), 0.17% Hepes, and 1% penicillin-streptomycin (Pen-Strep); filter-sterilized; and adjusted to pH 7.3. After dissection, the hippocampi were washed twice with cold HBSS and individually incubated at 37°C for 20 min in Papain dissociation

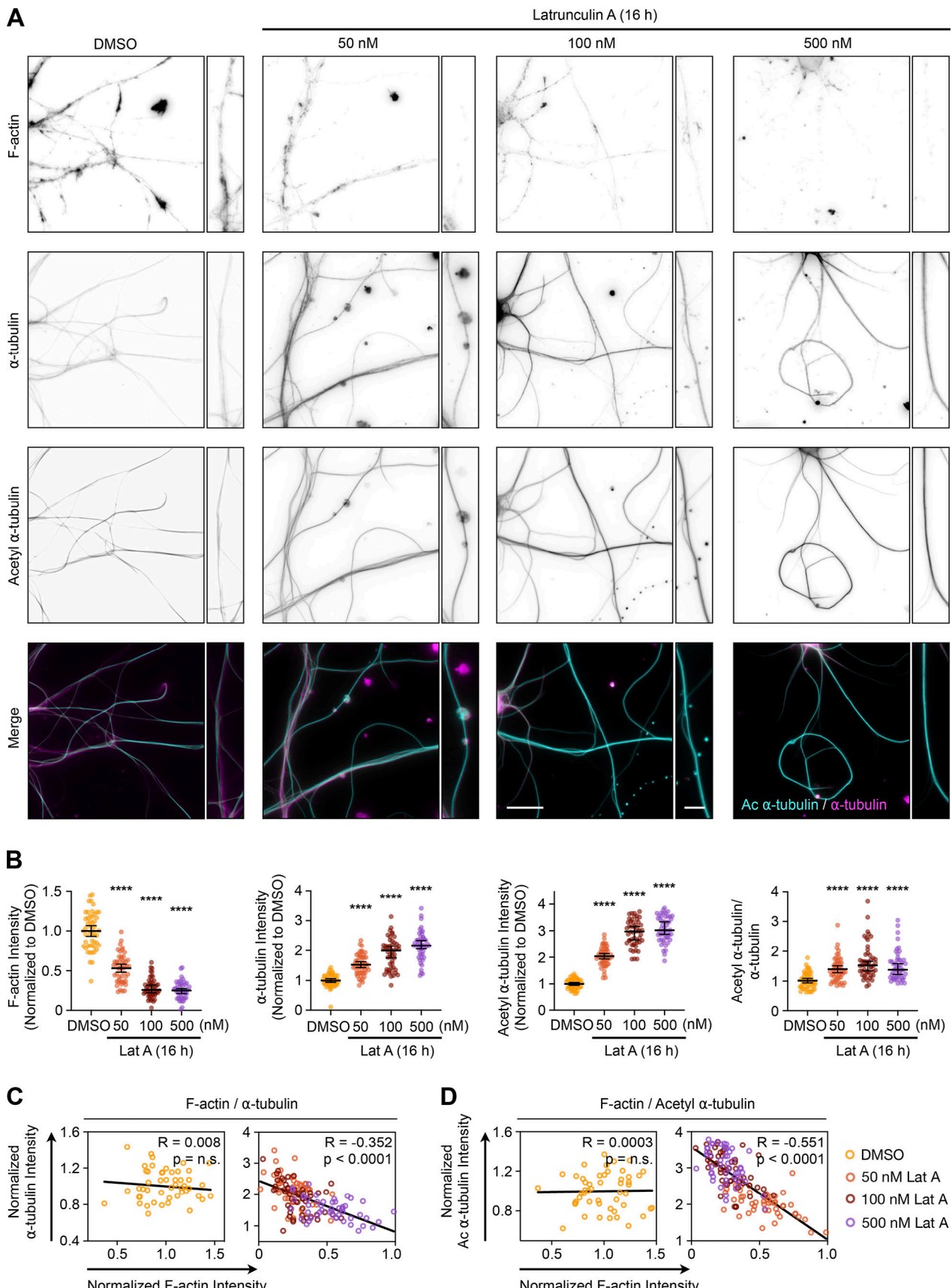

Figure 4. **Prolonged depletion of F-actin increases the number and acetylation of microtubules in the processes of hippocampal neurons. (A)** From top to bottom, representative images of F-actin, α-tubulin, acetyl α-tubulin, and merge images (acetyl α-tubulin: cyan; α-tubulin: magenta) in hippocampal

neurons incubated with 0, 50, 100, or 500 nM Latrunculin A (Lat A) for 16 h. Scale bar: 15 µm. Inset scale bar: 5 µm. **(B)** Quantification of the mean fluorescence intensities in A. From left to right, F-actin, α-tubulin, acetyl α-tubulin, and acetyl α-tubulin/α-tubulin ratio. Data are normalized to Ctrl (F-actin, α-tubulin, and acetyl α-tubulin) and plotted as mean ± 95% CI. n = 50 fields for each condition. Significance was calculated against control using ANOVA and Dunnett's post hoc test. **(C and D)** Correlation between fluorescence intensities for neurites in A. F-actin versus α-tubulin (C), and F-actin versus acetyl α-tubulin (D). Intensities were normalized to the mean of each dataset. **** indicates P < 0.0001, ns = not significant (P > 0.05).

solution 45 U of papain (Worthington), 0.01% deoxyribonuclease (DNase), 1 mg of DL-cysteine, 1 mg of bovine serum albumin (BSA), and 25 mg of D-glucose (all from Sigma-Aldrich) in phosphate-buffered saline (PBS). After digestion, the hippocampi were washed twice with DMEM, preheated to 37°C, supplemented with 10% FBS, and dissociated by 10 cycles of aspiration through a micropipette tip. Dissociated neurons were then resuspended in warm DMEM supplemented with 10% FBS and plated in six-well plates containing 25-mm sonicated glass coverslips pretreated with 50 µg/ml poly-L-lysine (PLL; Sigma-Aldrich). After 1 h, the medium was replaced with Neurobasal-A medium, which was supplemented with 2% B-27 and 0.25% GlutaMAX (neuronal medium). All reagents were from Gibco except when otherwise indicated. Primary neurons were maintained in a standard tissue culture incubator at 37°C with 5.5% $CO_2$.

### DNA transfection
CAD cells were transfected with plasmid DNA by electroporation using the Neon Transfection System (Invitrogen) and the Neon Transfection Kit (cat#MPK1096B; Invitrogen) as previously described (Skruber et al., 2020). Briefly, cells were grown to a confluency of 80%, trypsinized, and pelleted by centrifugation. Then, the pellet was rinsed twice with Dulbecco's Phosphate-Buffered Saline (DPBS, cat#21-031-CV; Corning) and resuspended in a minimum amount of buffer R (Neon Transfection Kit component) with 1 µg of DNA. Cells transfected with DNA constructs were cultured for 24–48 h. Before experiments were performed, cells were grown for 3 h on 10 µg/ml laminin-coated coverslips.

The following DNA constructs (Addgene) were used in this study: EGFP-C1(#54759; Plasmid) and mEGFP-PFN1 (#56438; Plasmid). Additional constructs used, such as mEGFP-PFN1-R88E (generated by us and described in Skruber et al. [2020]), PFN1-ALS mutants M114T, E117G, and G118V were generated from EGFP-PFN1 plasmid with site-directed mutagenesis (Q5 New England Biolabs) using the following primers: M114T: 5′-GTCCTGCTGACGGGCAAAGAAG-3′ (forward) and 5′-CCTTCT TTGCCCGTCAGCAGGAC-3′ (reverse), E117G: 5′-ATGGGCAAA GGAGGTGTCCAC-3′ (forward) and 5′-GGACACCTCCTTTGCCCA TC-3′ (reverse), G118V: 5′-ATGGGCAAAGAAGTTGTCCACGGT GGTTTG-3′ (forward) and 5′-CAAACCACC GTGGACAACTTC TTTGCCCAT-3′ (reverse). All constructs were prepared for transfection using either the GenElute HP Endotoxin-Free Plasmid Maxiprep Kit (Sigma-Aldrich) or the NucleoBond Xtra Midi EF kit (MACHEREY-NAGEL). Correct inserts were confirmed by sequencing (Genewiz).

### Western blotting
To prepare whole-cell lysates, we used a cell scraper to harvest cells in RIPA Lysis and Extraction Buffer (cat#89900; Thermo Fisher Scientific). We then passed the mixture through needles

of various gauges (21, 25, and 27) five times. Next, protein quantification was assessed with Pierce BCA Protein Assay Kit (cat#23227; Thermo Fisher Scientific) and diluted in SDS buffer stained with Orange G (40% glycerol, 6% SDS, 300 mM Tris HCl, 5% β-mercaptoethanol pH 6.8). The samples were then denatured at 95°C for 5 min before loading 10 µg of samples onto a SDS-PAGE gel (Novex 4%–20% Tris-Glycine Mini Gels, cat#XP04200BOX; Thermo Fisher Scientific). Proteins were transferred to a PVDF membrane 0.2 µm (cat#10600021; Amersham) and blocked in 5% Bovine Serum Albumin (BSA, cat#A9418; Sigma-Aldrich) for 20 min. All antibodies were diluted in 5% BSA and 0.1% Tween-20 (cat#J20605AP, Thermo Fisher Scientific). The following antibodies/dilutions were used: rabbit polyclonal anti-alpha Tubulin (1/500 dilution, cat#ab4074; Abcam), mouse monoclonal anti-alpha Tubulin (acetyl K40) (6-11B-1) (1/800 dilution, cat#ab24610; Abcam), rabbit monoclonal anti-GAPDH (14C10) (1/ 1,000, cat#2118S; Cell Signaling). For secondary antibodies, goat anti-mouse (1/10,000 dilution, 2 h at room temperature, cat#926-32210; Li-Cor) and goat anti-rabbit (1/10,000 dilution, 2 h at room temperature, cat#926-32211; Li-Cor) were used for imaging on the Li-Cor Odyssey detection system.

### Actin depolymerization
To evaluate the effect of actin depolymerization on microtubules, we used Latrunculin A (Lat A, cat#BML-T119-0100; Enzo Life Sciences) dissolved in DMSO (cat#D2650; Sigma-Aldrich) to treat CAD cells and primary hippocampal neurons prior to immunostaining and immunoblotting. For imaging of Lat A–treated CAD cells, cells were grown on laminin-coated coverslips for 3 h and then treated with 10 nM Lat A for 16 h at 37°C in a standard tissue culture incubator before immunostaining for imaging and analysis. Hippocampal primary neurons were seeded onto PLL-coated coverslips and cultured for 3 days before being treated with either 50, 100, or 500 nM Lat A for 24 h at 37°C. For Western blotting of CAD cells, cells cultured in 100-mm dishes to 80–90% confluence were treated with 10 nM Lat A for 16 h at 37°C before harvesting for protein electrophoresis.

### Reducing and increasing actomyosin contractility
To evaluate the effect of actomyosin contractility on microtubules, we inhibited and hyperactivated non-muscle myosin II ATPase with (–)-Blebbistatin (cat#20-339-11MG; MilliporeSigma Calbiochem) and Calyculin A (cat# C5552; Sigma-Aldrich), respectively. For Blebbistatin experiments, hippocampal neurons were first seeded onto PLL-coated coverslips and cultured for 3 days, then treated with 15 µM Blebbistatin for 24 h at 37°C, after which the medium was replaced with fresh neuronal medium for 24 h before performing immunocytochemistry, imaging, and analysis as described above. CAD cells were cultured in 100-mm dishes at 80–90% confluence and then treated with 15 µM

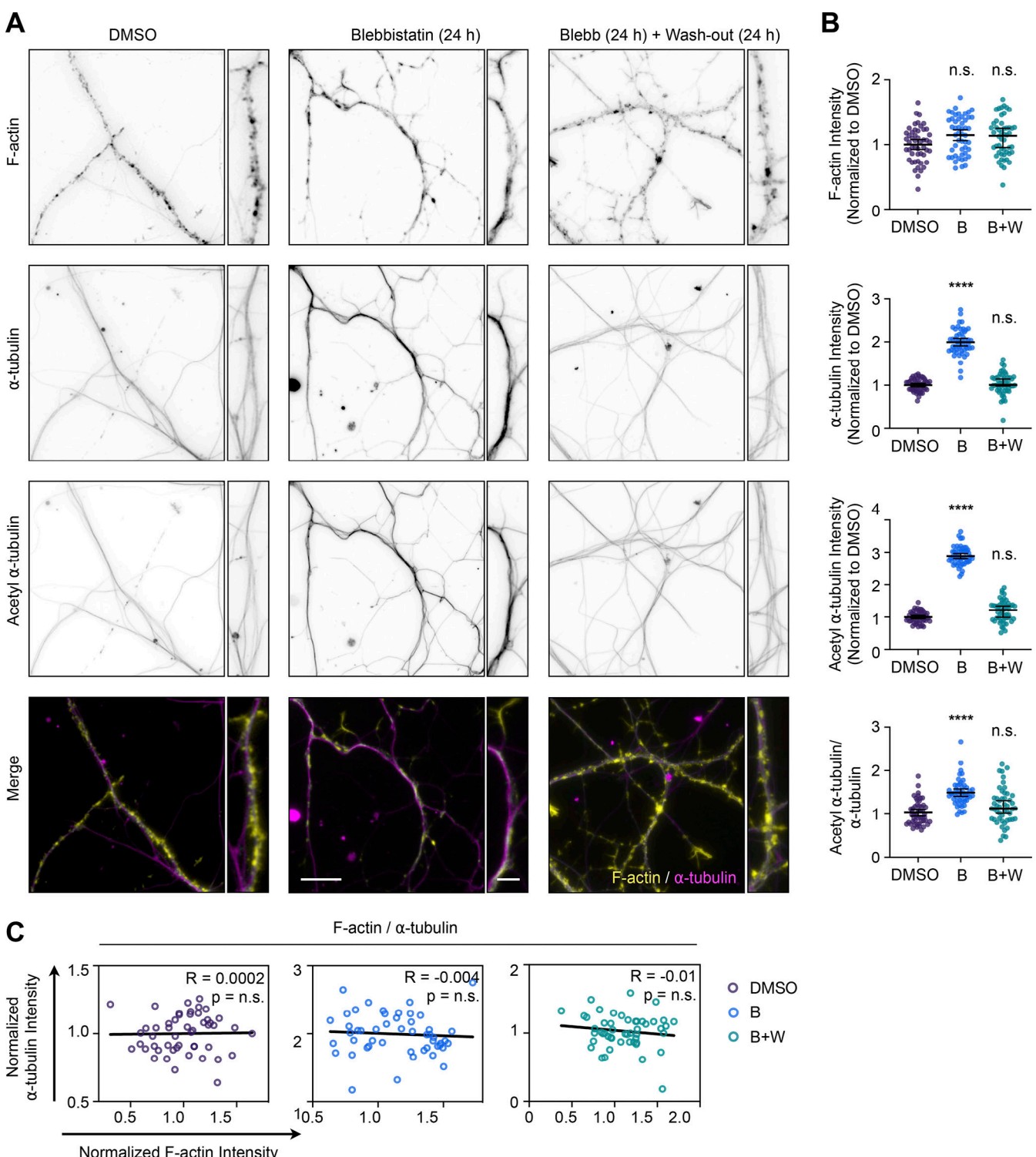

Figure 5. **Prolonged inhibition of actomyosin contractility increases the number and acetylation of microtubules without depolymerizing F-actin in hippocampal neuron processes. (A)** From top to bottom, representative images of F-actin, α-tubulin, acetyl α-tubulin, and merge images (F-actin: yellow; α-tubulin: magenta) in hippocampal neurons incubated with 0, 15 μM Blebbistatin (Blebb) for 24 h, or Blebb for 24 h and then washed and incubated with fresh medium for 24 h. Scale bar: 15 μm. Inset scale bar: 5 μm. **(B)** Quantification of the mean fluorescence intensities in A. From top to bottom, F-actin, α-tubulin, acetyl α-tubulin, and acetyl α-tubulin/α-tubulin ratio. Data are normalized to Ctrl (F-actin, α-tubulin, and acetyl α-tubulin) and plotted as mean ± 95% CI. n = 50 fields for control and Blebb (24 h) + Wash-out (24 h) (B+W), n = 53 fields for Blebbistatin for 24 h (B). Significance was calculated against control using ANOVA and Dunnett's post hoc test. **(C)** Correlations between F-actin and α-tubulin intensities for neurites in A. Intensities were normalized to the mean of each dataset. **** indicates P < 0.0001, ns = not significant (P > 0.05).

Blebbistatin for 24 h at 37°C after which the medium was replaced with fresh complete medium for 24 h before harvesting cells for Western blotting. For Calyculin A experiments, CAD cells were grown on laminin-coated coverslips for 3 h and then treated with 5 nM Calyculin A for 16 h at 37°C in a standard tissue culture incubator before immunostaining for imaging and analysis.

## Immunostaining

Cells were fixed and permeabilized with 3% paraformaldehyde (Ca#15710; Electron Microscopy Sciences), 0.1% glutaraldehyde (Ca#16019; Electron Microscopy Sciences), 4% sucrose (catC12H22O11; Fisher Chemical), 0.5% Triton X-100 (cat#BP151; Fisher Bioreagents), 0.1 M 1,4-Piperazinediethanesulfonic acid (PIPES, cat#4265-01; J.T. Baker) pH 7.0, 2 mM ethylene glycol-bis(2-aminoethylether)-N,N,N′,N′-tetraacetic acid (EGTA, cat#428570100; Acros Organic), and 2 mM magnesium chloride (cat#BP214; Thermo Fisher Scientific) for 10 min at room temperature (RT). Cells were then washed three times with PBS at RT and incubated with primary antibody at 37°C for 2 h. They were then washed four times with PBS at RT and incubated with secondary antibody for 1 h at RT, then washed four times with PBS at RT. Actin filaments were stained with Alexa Fluor 488 phalloidin, Alexa Fluor 568 phalloidin, or Alexa Fluor 647 phalloidin (1/100 dilutions, cat#A12379; A12380; A22287; respectably; Invitrogen) in PBS for 30 min at RT. Cells were washed four times with PBS before mounting with ProLong Diamond Antifade Mountant (cat#P36961; Invitrogen) or with ProLong Diamond Antifade Mountant with DAPI (cat#P36962; Invitrogen). The following primary antibodies were used: rabbit polyclonal anti-alpha Tubulin (1/500 dilution, cat#ab4074; Abcam), mouse monoclonal anti-alpha Tubulin (acetyl K40) (6-11B-1) (1/800 dilution, cat#ab24610, clone 6-11B-1; Abcam), rabbit polyclonal anti-beta Tubulin (1/500 dilution, cat#ab6046; Abcam), and mouse monoclonal anti-TOM20 (4F3) (1/500 dilution, cat#ab56783; Abcam). The following secondary antibodies were used: goat anti-mouse 488 (1/500 dilution, cat#A11029; Invitrogen), goat anti-rabbit 568 (1/500 dilution, cat#A11011; Invitrogen), and donkey anti-rabbit 488 (1/500 dilution, cat#A32790; Invitrogen).

MEFs and CAD cells were seeded onto coverslips coated with 10 µg/ml fibronectin (cat#356008; Corning) or 10 µg/ml laminin (cat#L2020; Sigma-Aldrich), respectively, and cultured for 3 h before fixing and permeabilization process. To perform immunostaining in hippocampal neurons, they were seeded onto coverslips coated with 50 µg/ml PLL and cultured for 3 days before the experiments.

## Microscopy

Imaging was performed using an EVOS M5000 digital inverted microscope (Life Technologies) equipped with Olympus UP-PlanXApo 20X 0.8NA and 40X 0.95NA objectives and an integrated 3.2 MP monochrome CMOS camera; a Nikon A1R+ Laser Scanning Confocal Microscope equipped with Apo TIRF 60X 1.49 N.A. objective, and a GaAsP multidetector unit; or a Nikon CSU-W1 SoRa Spinning Disk Confocal Microscope using a 100X 1.49NA SR objective and a Hamamatsu Fusion BT Camera. Images from the

EVOS M5000 were acquired using the integrated EVOS imaging software. Images from Nikon microscopes were acquired using NIS-Elements (Nikon) software. All data from the SoRa microscope used the SoRa spinning disk unit, had background noise removed using Denoise.ai (NIS-Elements; Nikon), and was deconvolved using the Blind algorithm (10 or 15 iterations, with spherical aberration correction) in NIS-Elements.

## Mitochondria and lysosome live-cell imaging

Mitotracker Red and Lysotracker Green (Thermo Fisher Scientific) were used at 50 nM concentration for live cell imaging. Briefly, differentiated CAD cells grown on laminin-coated coverslips for 4 days were removed from the incubator, washed once in DPBS (Gibco), and stained for 30 min at 37°C with 50 nM Mitotracker or Lysotracker (dissolved in DMSO and diluted in cell culture medium without serum). After staining, cells were washed twice in a complete cell culture medium and mounted in imaging chambers with imaging media (serum- and phenol red–free DMEM/F12 with 20 mM HEPES). Mitochondria and lysosomes were imaged using the Nikon A1R+ Laser Scanning Confocal Microscope. Movies were analyzed in Fiji/ImageJ using Trackmate (Tinevez et al., 2017).

## Quantification of actin and microtubules from microscope images

To measure F-actin, tubulin, or acetyl α-tubulin from microscope images, samples were prepared for microscopy and imaged as described above. Images were then exported into Fiji/ImageJ and background-subtracted. Undifferentiated cells were traced using the F-actin channel with the magic wand tool. This region of interest was then transferred to other channels so that the mean intensity within the region could be recorded. For differentiated CAD cells or neurons, the measurements were made using a 10 × 30 µm rectangle as the region of interest in processes to similar distances from the cell body. For image quantification experiments listed above, results were obtained from three independent biological replicates. We quantified filopodia in differentiated CAD cells as previously described (Osking et al., 2019). Briefly, 3D confocal z-stacks from a laser-scanning confocal microscope were deconvolved with NIS-Elements software using the Landweber algorithm (15 iterations, with spherical aberration correction). The ImageJ plugin Filopodyan (Urbančič et al., 2017) was used to segment and quantify individual filopodia using the phalloidin channel on the deconvolved, maximum-intensity projection images.

## Data analysis and statistics

Unless noted, all data were obtained using three independent biological replicates. All data were tested for normality using the Shapiro–Wilk normality test. If the data assumed a Gaussian distribution, groups were compared using either an unpaired two-sided Student's $t$ test for two conditions or an ordinary one-way ANOVA for three or more conditions. ANOVA was followed by Dunnett's post-hoc test for comparisons of all conditions against the control condition or by Tukey's post-hoc test for comparisons of all conditions with each other. If the data failed the normality test, then two groups were compared using the

Mann–Whitney test. Results are presented normalized to control as mean ± 95% confidence interval (CI). Analysis and graphing of results were performed using GraphPad Prism 10 software.

### Online supplemental material

Fig. S1 shows that depleting PFN1 in mouse embryonic fibroblasts has the same effect on microtubules as knocking out PFN1 in CAD cells. Fig. S2 shows that undifferentiated and differentiated PFN1 KO CAD cells have elevated levels of neurofilament heavy chain. Fig. S3 uses pharmacological inhibitors to show that manipulating myosin activity has no effect on tubulin acetylation in PFN1 KO or F-actin depleted CAD cells.

### Data availability

The data that support the findings of this study are available upon reasonable request from the corresponding author (Eric A. Vitriol).

## Acknowledgments

We would like to thank Francesca Bartolini (Columbia University) for asking the question that inspired this study and attendees of the Chicago and Triangle Cytoskeleton Meetings for helpful discussions and feedback about this work.

The research reported in this publication was supported by the Maximizing Investigators' Research Award from the National Institute of General Medical Sciences of the National Institutes of Health under grant number R35GM137959 to E.A. Vitriol.

Author contributions: B.A. Cisterna: Conceptualization, Data curation, Formal analysis, Investigation, Methodology, Project administration, Validation, Visualization, Writing—original draft, and Writing—review and editing; K. Skruber: Investigation; M.L. Jane: Data curation and Formal analysis; C.I. Camesi: Formal analysis and Investigation; I. D. Nguyen: Data curation, Formal analysis, and Investigation; T.M. Liu: Visualization; P.V. Warp: Data curation and Investigation; J.B. Black: Conceptualization, Data curation, and Resources; M.T. Butler: Resources; J.E. Bear: Funding acquisition and Resources; D.E. Mor: Investigation; T.-A. Read: Conceptualization, Investigation, Methodology, Supervision, Writing—original draft, and Writing—review and editing; E.A. Vitriol: Conceptualization, Data curation, Formal analysis, Funding acquisition, Investigation, Methodology, Project administration, Resources, Supervision, Visualization, Writing—original draft, and Writing—review and editing.

Disclosures: The authors declare no competing interests exist.

Submitted: 19 September 2023

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

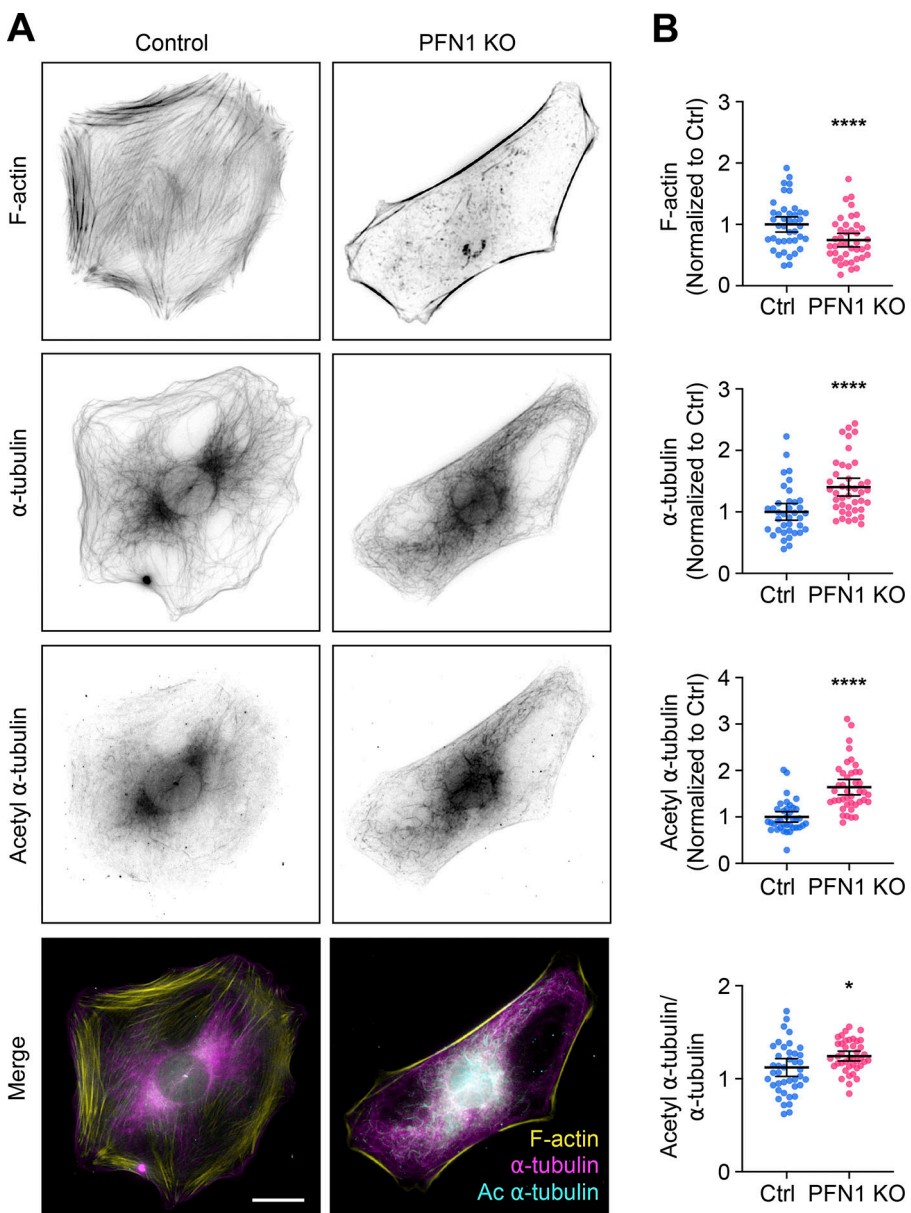

Figure S1. **Knocking out PFN1 in mouse embryonic fibroblasts increases the number and acetylation of microtubules. (A)** From top to bottom, representative images of F-actin, α-tubulin, acetyl α-tubulin, and merge (F-actin: yellow; α-tubulin: magenta; acetyl α-tubulin: cyan) in Control and PFN1 KO mouse embryonic fibroblasts (MEFs) cells. Scale bar: 10 μm. **(B)** Quantification of the mean fluorescence intensities in A. From top to bottom, F-actin, α-tubulin, acetyl α-tubulin, and acetyl α-tubulin/α-tubulin ratio. Data are normalized to Ctrl (F-actin, α-tubulin, and acetyl α-tubulin) and plotted as mean ± 95% CI. $n$ = 41 cells for each condition. Significance was calculated using Student's $t$ test. **** indicates $P < 0.0001$, * indicates $P < 0.05$.

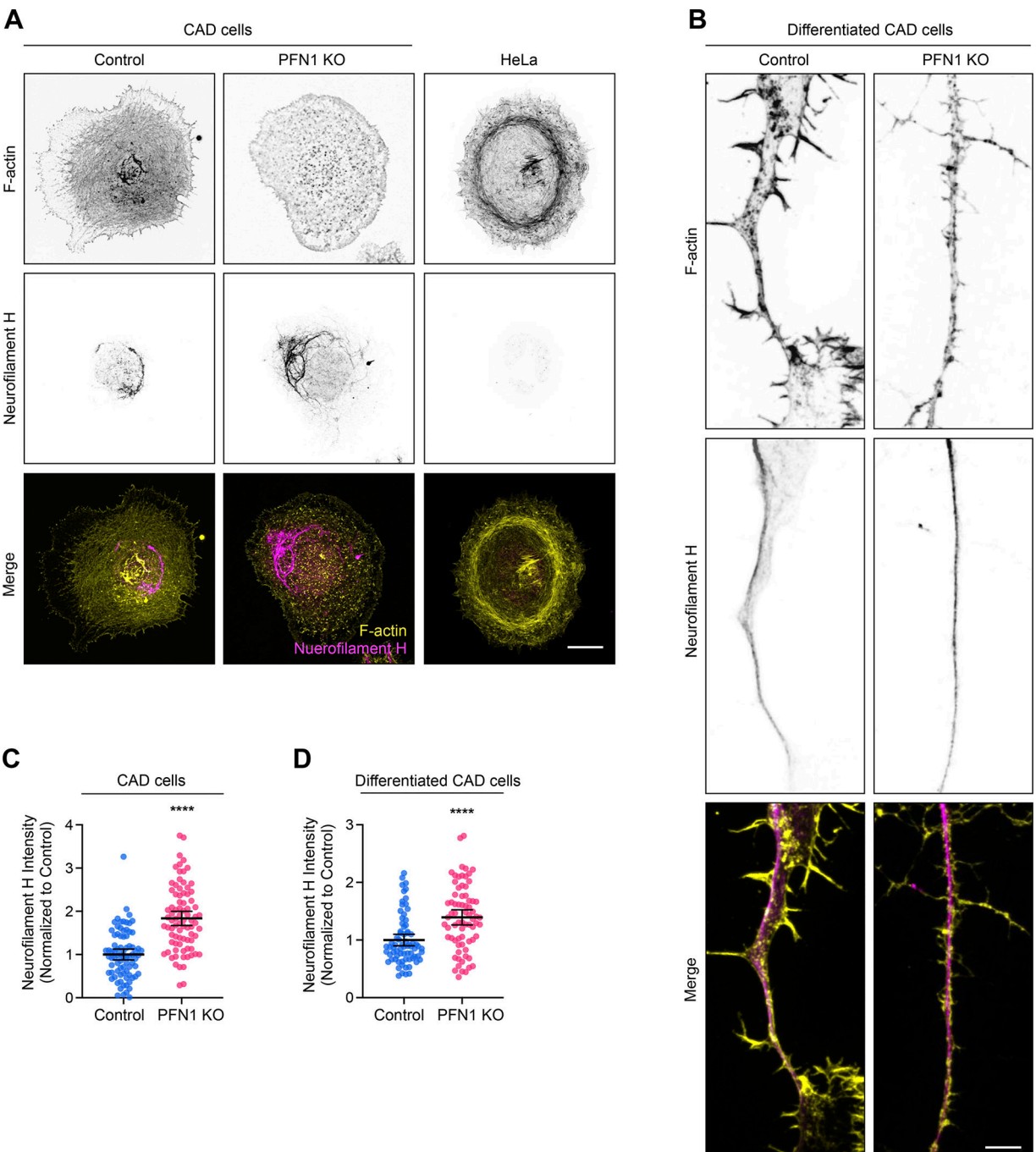

Figure S2. **PFN1 KO CAD cells have an elevated expression of neurofilament heavy chain. (A)** Representative images of F-actin at the top, neurofilament heavy chain (Neurofilament H) in the middle, and merge images (F-actin: yellow; Neurofilament H: magenta) at the bottom in control, PFN1 KO CAD cells, and HeLa cells, which were used a non-neuronal control. Scale bar: 10 μm. **(B)** Representative images of F-actin, Neurofilament H, and merge images in the processes of differentiated Control and PFN1 KO CAD cells. Scale bar: 4 μm. **(C)** Quantification of mean fluorescence intensities of cells in A. Data are normalized to control and plotted as mean ± 95% CI. $n$ = 80 cells for control and $n$ = 82 cells for PFN1 KO. Significance was calculated using Student's $t$ test. **(D)** Quantification of mean fluorescence intensities of neuron-like processes of differentiated CAD cells in B. Data are normalized to control and plotted as mean ± 95% CI. $n$ = 73 processes for control and PFN1 KO. Four independent experiments Significance was calculated using Student's $t$ test. **** indicates $P < 0.0001$.

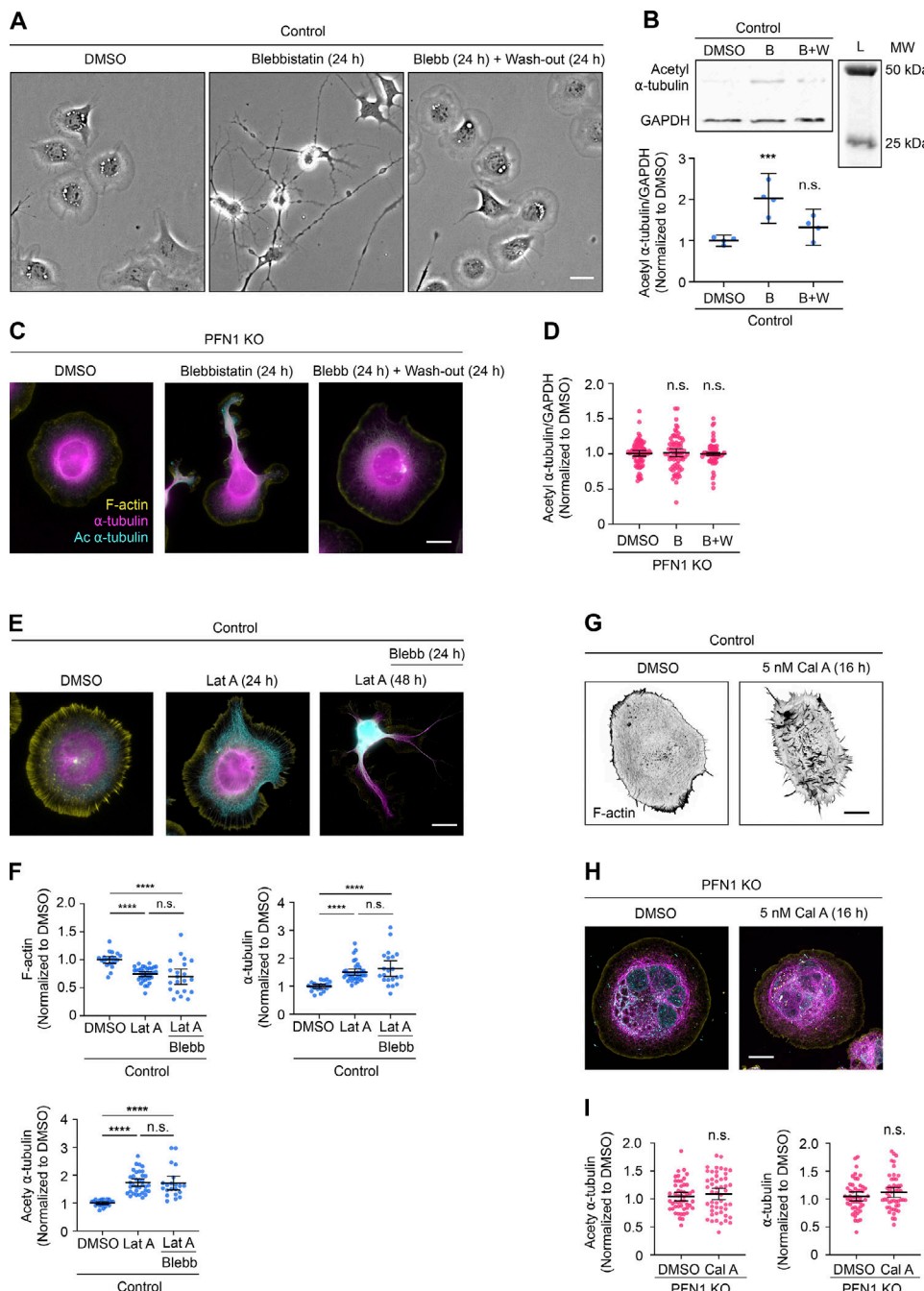

Figure S3. **Manipulating myosin activity has no effect on tubulin acetylation in PFN1 KO or F-actin depleted CAD cells. (A)** Representative bright field images of CAD cells incubated with 0, 15 µM Blebbistatin (Blebb) for 24 h, or Blebb for 24 h and then washed and incubated with fresh medium for 24 h. Scale bar: 20 µm. **(B)** Western blot of acetyl α-tubulin and GAPDH in CAD cells incubated with 0, 15 µM Blebb for 24 h (B), or Blebb for 24 h and then washed and incubated with fresh medium for 24 h (B+W) at the top, and quantification of levels expression at the bottom. Data are normalized to control and plotted as mean ± 95% CI. Four independent experiments. Significance was calculated against control using ANOVA and Dunnett's post hoc test. **(C)** Representative merge images of F-actin (yellow), α-tubulin (magenta), and acetyl α-tubulin (cyan) in PFN1 KO CAD cells incubated with 0, 15 µM Blebb for 24 h, or Blebb for 24 h and then washed and incubated with fresh medium for 24 h. Scale bar: 10 µm. **(D)** Quantification of mean acetyl α-tubulin intensities in C. Data are normalized to Ctrl and plotted as mean ± 95% CI. n = 80 cells for PFN1 KO, n = 72 cells for Blebb (B), and n = 60 cells for B+W. Significance was calculated against control using ANOVA and Dunnett's post hoc test. **(E)** Representative merge images of F-actin, α-tubulin, acetyl α-tubulin in CAD cells incubated with 0, 10 nM Lat A for 24 h, or cells incubated with Lat A for 48 h where Blebb was added after the first 24 h. Scale bar: 10 µm. **(F)** Quantification of mean fluorescence intensities in E. F-actin and α-tubulin at the top, and acetyl α-tubulin in the bottom. Data are normalized to Ctrl and plotted as mean ± 95% CI. n = 21 cells for Control and Lat A plus Blebb, and n = 35 cells for Lat A. Significance was calculated using ANOVA and Tukey's post hoc test. **(G)** Representative images of F-actin in CAD cells incubated with 0 or 5 nM Calyculin A (Cal A) for 16 h. Scale bar: 10 µm. **(H)** Representative merge images of F-actin, α-tubulin, and acetyl α-tubulin in PFN1 KO CAD cells incubated with 0 or 5 nM Cal A for 16 h. Scale bar: 10 µm. **(I)** Quantification of mean α-tubulin and acetyl α-tubulin intensities in H. Data are normalized to control and plotted as mean ± 95% CI. n = 50 cells for all conditions. Significance was calculated using Student's t test. **** indicates P < 0.0001, ns = not significant (P > 0.05). Source data are available for this figure: SourceData FS3.

