## [Peer Review File · The Journal of Cell Biology]

Prolonged depletion of profilin 1 or F-actin causes an adaptive response in microtubules

Bruno Cisterna, Kristen Skrubber, Makenzie Jane, Caleb Gimesi, Ivan Nguyen, Tatiana Liu, Peyton Warp, Joseph Black, Mitchell Butler, James Bear, Danielle Mor, Tracy-Ann Read, and Eric Vitriol

Corresponding Author(s): Eric Vitriol, Augusta University and Tracy-Ann Read, Augusta University

Review Timeline:

Submission Date:	2023-09-19
Editorial Decision:	2023-11-01
Revision Received:	2024-03-06
Editorial Decision:	2024-03-12
Revision Received:	2024-04-05

Monitoring Editor: Kenneth Yamada

Scientific Editor: Tim Spencer

Transaction Report:

DOI: <https://doi.org/10.1083/jcb.202309097>

November 1, 2023

Re: JCB manuscript #202309097

Dr. Eric Vitriol
Augusta University
1462 Laney Walker Blvd.
Augusta 30912

Dear Dr. Vitriol,

Thank you for submitting your manuscript entitled "Profilin 1 predominantly regulates microtubules through actin in cells". The manuscript has now been assessed by three expert reviewers, whose reports are appended below. As you can see from the reviews provided by three leaders in various overlapping research areas spanning the elements of this paper, there was significant interest in the conclusions, but also some concerns and suggestions for strengthening the manuscripts. We invite you to submit a revision if you can address the reviewers' key concerns, as outlined here.

The reviewers addressed different aspects of the manuscript, but all three raised useful specific points that we ask you to address as directly as possible. For example, additional evidence for the effect of actomyosin restoration and ensuring that the text wording is accurate will be helpful, though we do not feel that you will need to alter Figures 1 and 2 unless you wish to do so. We do feel that many of the points raised by these conscientious reviewers can be resolved in straightforward fashion, either by the addition of limited amounts of additional or stronger data, or by clarifications within the text. Please address directly each specific comment in point-by-point fashion, adding whatever additional experimental data will be practical provide within the next couple of months.

We look forward to receiving a resubmitted manuscript from you, which will be re-evaluated by us with input from at least one of the original reviewers.

GENERAL GUIDELINES:

Text limits: Character count for a Report is < 20,000, not including spaces. Count includes title page, abstract, introduction, the joint Results & Discussion, and acknowledgments. Count does not include materials and methods, figure legends, references, tables, or supplemental legends.

Figures: Reports may have up to 5 main text figures. To avoid delays in production, figures must be prepared according to the policies outlined in our Instructions to Authors, under Data Presentation, <https://jcb.rupress.org/site/misc/ifora.xhtml>. All figures in accepted manuscripts will be screened prior to publication.

*****IMPORTANT:** It is JCB policy that if requested, original data images must be made available. Failure to provide original images upon request will result in unavoidable delays in publication. Please ensure that you have access to all original microscopy and blot data images before submitting your revision. *******

Supplemental information: There are strict limits on the allowable amount of supplemental data. Reports may have up to 3 supplemental figures. Up to 10 supplemental videos or flash animations are allowed. A summary of all supplemental material should appear at the end of the Materials and methods section.

Please note that JCB now requires authors to submit Source Data used to generate figures containing gels and Western blots with all revised manuscripts. This Source Data consists of fully uncropped and unprocessed images for each gel/blot displayed in the main and supplemental figures. Since your paper includes cropped gel and/or blot images, please be sure to provide one Source Data file for each figure that contains gels and/or blots along with your revised manuscript files. File names for Source Data figures should be alphanumeric without any spaces or special characters (i.e., SourceDataF#, where F# refers to the associated main figure number or SourceDataFS# for those associated with Supplementary figures). The lanes of the gels/blots should be labeled as they are in the associated figure, the place where cropping was applied should be marked (with a box), and molecular weight/size standards should be labeled wherever possible.

Source Data Figures should be provided as individual PDF files (one file per figure). Authors should endeavor to retain a minimum resolution of 300 dpi or pixels per inch. Please review our instructions for export from Photoshop, Illustrator, and

PowerPoint here: <https://rupress.org/jcb/pages/submission-guidelines#revised>

The typical timeframe for revisions is three to four months. Please note that papers are generally considered through only one revision cycle, so any revised manuscript will likely be either accepted or rejected.

Thank you for this interesting contribution to Journal of Cell Biology. You can contact us at the journal office with any questions at cellbio@rockefeller.edu.

Sincerely,

Kenneth M. Yamada, MD, PhD
Senior Editor
Journal of Cell Biology

Tim Spencer, PhD
Executive Editor
Journal of Cell Biology

Reviewer #1 (Comments to the Authors (Required)):

Recent studies provided evidence that a central regulator of actin dynamics, profilin, also binds microtubules and alters their polymerization *in vitro*. Moreover, profilin depletion/mutations are associated with abnormal phenotypes of microtubules in cells. However, whether profilin regulates microtubules directly, or indirectly through the actin cytoskeleton has remained elusive. Here, Cisterna et al. approached this question by knockout-rescue experiments with different profilin mutants and by analyzing the effects of actin and myosin inhibitors on microtubules. Importantly, they show that in undifferentiated and differentiated (neuron-like) CAD cells, as well as in primary neurons, profilin depletion (or disrupting the actin cytoskeleton by long-term treatment with Latrunculin A) results in increased α -tubulin intensity and acetylation, and provide evidence that these effects of profilin depletion on microtubules occur primarily through the actin cytoskeleton.

The data presented in the manuscript appear of good technical quality, and the study provides important new information on the mechanisms by which profilin controls microtubules in cells. This work also elucidates the mechanisms by which certain profilin mutations contribute to ALS. However, there are few points that should be addressed to strengthen the study.

1. The authors state in the Abstract that 'All the observed changes to microtubules were reversible if actomyosin was restored, arguing that PFN1's regulation of microtubules occurs principally through actin'. This is not entirely correct, because this conclusion is only based on bebbistatin washout experiments. Thus, if the authors wish to state that profilin's regulation of microtubules occurs mainly through actin, they should examine if the MT-phenotype of PFY1 knockout cells could be (partially) rescued either by elevating the levels of cytoplasmic actin (see e.g. Suarez et al., *Dev. Cell.* 2015), or by increasing the contractility e.g. by expressing dominant active RhoA in the knockout cells.
2. It would be informative to show representative examples of the actin phenotypes of different PFY1 knockout-rescue cells (with different mutant profilins) as an additional supplementary figure. This is important, because apart from the effects of these profilin mutants on F-actin levels, it remained unclear if these mutants have distinct effects on lamellipodial actin filaments, microspikes, or contractile actomyosin bundles.
3. The authors should clarify in the legend to Fig. 1D that 'control cells' are PFY1 knockout cells, not wild-type cells (I assume that this was the case).
4. Fig. 2: Links between the actin cytoskeleton and MT-phenotype in the differentiated CAD cells remain unclear, unless the authors will carry out LatA treatment (or rescue experiments with the profilin R88E mutant) also in the differentiated CAD cells.

Reviewer #2 (Comments to the Authors (Required)):

Actin-microtubule crosstalk is an area of emerging interest. The exact molecular mechanisms by which these two cytoskeletal

networks exert control over each other, however, still remain largely unresolved. One such mechanism which has recently received attention is the direct binding of ubiquitous actin-binding protein profilin to microtubules.

While profilin has previously been shown to regulate microtubule dynamics in vitro, in vivo implications of this interaction are less clear. The current manuscript shows that in cells, profilin's effects on microtubules are not due to its direct binding to microtubules as has been previously suggested (Henty- Ridilla et al, 2017). The authors here show that profilin-related changes in microtubule dynamics are indirect and mediated by profilin's effects on actin assembly and actomyosin contractility rather than profilin's direct interactions with microtubules. More specifically, they report that these effects of profilin are enhanced in neuronal processes and that they mimic changes in actin cytoskeletal found in neurodegenerative diseases. Authors convincingly show that their results are consistent across multiple cell lines, differentiated and undifferentiated, and provide insights into how profilin's ALS mutants can affect progenitor cells vs neuronal-like cells.

Data appears to be carefully acquired, rigorously analyzed and presented well. Although the manuscript is well-written overall, I think certain parts can be expanded for clarity (see below). While I don't have any major concerns about the manuscript, I mention below a few minor points which I hope will help improve the manuscript.

1. In general, I found both the text and figures to be very dense. I believe this might be a result of the limitations imposed by the report format. As an example, some of the panels in Figure 1 could be moved to supplementary data. Authors could also consider splitting Figure 2 into two separate figures - panels A-H focusing on quantification in knockout vs control, and panels I-M focusing on the effects of perturbations.
2. Figure 2D - I am not sure if the image below the graph is helpful since I didn't find any mention of it in the text.
3. Figure 3 - What was the reason for looking only into the heavy chain subunit? An argument for why authors decided to focus only on the heavy chain would be helpful for non-expert readers.
4. Figure 3B- I didn't find any mention of HeLa cells in the main text. I assume they were used as an example of non-neuronal cells. If it is the case, explicitly mentioning this in the text might help.
5. Figure 3 - The text in the manuscript describing the figure could be expanded.
6. In the results section talking about Figure 4- 'neurites' needs to be deleted from the first sentence.
7. Figure 4B- Images of α -tubulin and acetyl α -tubulin are mis-aligned i.e. they are not exactly the same fields of view. I assume that the authors have done all the quantification on images that were properly aligned, but still worth a check.
8. Figure 5A - Images of F-actin (top row) and the overlay (lower row) for both control and Blebbistatin treatment are mis-aligned. As mentioned above, they are not the same exact fields of view. Please correct.
9. I didn't find any mention of supplementary figure 2 in the main text.

Reviewer #3 (Comments to the Authors (Required)):

Overall, this is a manuscript of interest to the JCB readership. However, before it can be published, there are a number of points that I would like to raise and which should be considered by the authors to improve their work.

Firstly, as explained in my detailed comments, the paper requires a better argumentation which closely describes existing knowledge of PFN1 and then carefully argues as to whether data presented here provide potential alternative explanations or might even override existing views.

Second, in my view, the heavy reliance on CAD cells weakens the work. What do we know about these CAD cells and their neuritic actin organisation? Do they have the typical actin-spectrin sleeve? Does PFN1 loss affect that sleeve? The clumpy appearance upon PFN1 KO is unexpected and resembles drug-treated axons rather than a system that merely reduces its polymerisation efficiency. As explained in my detailed comments, there are some contradictions regarding axon width when imaged for actin versus microtubules, or when comparing CAD cells to hippocampal neurons. These contradictions should be carefully addressed.

Third, throughout the manuscript, conclusions need to be more precise (see detailed comments). For example, in contrast to what was concluded, LatA-mediated actin removal has been shown in CAD and hippocampal cells to influence tubulin amounts and acetylation, but none of the other parameters has been tested upon LatA application. Conclusions need to be carefully adapted to reflect the actual data, or experiments should be provided to support the current conclusions.

Fourth, it would have been helpful to see some readouts at growth cones where relationships between actin and microtubules

are far better understood; also the impact of PFN1 on parameters such as filopodia length or numbers would have helped. Existing images could be re-analysed with this in mind?

DETAILED COMMENTS

p.1, right: Please, explain the conventional function of Profilin before delving into microtubules.

p.1, right: A bit more background information on the findings why people believe Pfn1 is a microtubule regulator should be explained, so that the experiments shown here can be held against previous arguments.

p.1, left: Please, name a few other cross-linker far better studied in neurons: spectraplakins (e.g. <http://doi.org/10.1016/j.semcdb.2017.05.019>) and Drebrin/Eb3 (e.g. <https://doi.org/10.1038/ncb1778>; <https://doi.org/10.1111/jnc.15502>)

p.3, left: I do not understand the sentence 'Similarly, only WT and E117G expressing PFN1 KO cells lost the correlation between the amount of F-actin and microtubules.' given that they restore towards wildtype situation which is at one end of the correlation.

p.3 left: "while the causative ALS-linked PFN1 mutations have been shown to alter actin assembly in complex ways" - please, explain better. It is not clear what you mean: are different mutants behave different in the same essays, or are the same mutants behaving different in different essays?

Fig.2 I-M: Please, differentiate into antero- and retrograde transport to see whether dyneins and kinesins are similarly changed in their behaviours.

p.5, right: '~400% increase in number of MTs"; authors measured intensity of tubulin staining and should stick to this readout. They can argue that this likely reflects an increase in numbers and explain the approach they took in Fig.2D; readers should be able to decide as to whether they agree with this calculation. I don't, as also argued later.

p.5, right: "This highlighted the importance that morphology has in the response to cytoskeletal dysregulation." - I do not understand what the authors want to say. Please, reformulate.

p.5, right, 2nd para: Reformulate the first sentence along the lines of: "The maintenance of neurites requires the antero- and retrograde transport of materials ..."

p.5, left: The fact that neurites become narrower makes me question whether these are really neuron-like cells. For example, has it been shown that they have a periodic actin-spectrin cortex? It is confusing that the microtubules show the opposite trend, i.e. become thicker, as quantified in Fig.2D. Hippocampal neurons shown later on show a thinning of MT bundles. Intensity readings for CAD and hippocampal neurons become therefore difficult to compare and interpret. Something here does not look right.

Fig.3C,D: explain how measurements were done. Did you measure ROIs or entire cells or neurites? The 'D' label is missing from the Figure.

p.6, left, conclusion first para: This conclusion cannot be made since the authors used PFD1 KO rather than direct depletion of actin. PFD1 function may directly target MTs in neurites, especially when considering that there are functional differences between undifferentiated and differentiated CAD cells as described by the authors. --- Hippocampal cell experiments could be moved up here to make the point about MTs and acetylation stronger, whereas none of the other parameters (transport, NFs) were ever assessed upon LatA application.

p.6, left, 2nd para: Please, explain that these data are not from WB analyses but staining intensities. Explain what exactly was measured (info is also not provided in image), so that the statement 'no process-specific increase' can be understood.

p.6, right, middle. Please, indicate the readouts (i.e. WB and intensity) used to make your statements.

p.6, left, end of 2nd para: 'neurofilament accumulations' - this statement is not necessarily correct since the authors do not find pathological aggregates in their cells but proper filaments.

p.6, right: The last conclusion is vague; I would want to see a clear discussion as to how the results for PFN1 found here, relate to or override previous PFN1 models of direct MT regulation.

Fig.4B: The images in different channels are not well aligned, i.e. do not show the exact same position. In contrast to differentiated CA cells, the axons in hippocampal neurons seem to become thinner, i.e. MTs are potentially compacted which might explain the increased staining intensity (rather than an increase in MT number). The image for 500nm LatA application shows the extremely weak tubulin staining on the left, yet acetylation does not reflect this at all. I do not trust that image.

Please, indicate the identity of the rabbit anti- α -tub antibody.

Response to Reviewers

We would like to thank the reviewers for their fast and fair evaluation of our manuscript. Below is our point-by-point response, with our comments written in blue. In response to the constructive feedback we received, we have performed new experiments and revised the figures and text. As a result, the conclusions from the original manuscript have been strengthened, the presentation of results has been improved, and the language used to describe our findings has been made more precise. Thus, we believe the paper has been significantly improved through peer review. We hope that you agree and now find the manuscript suitable for publication in *JCB*.

Reviewer #1

Recent studies provided evidence that a central regulator of actin dynamics, profilin, also binds microtubules and alters their polymerization in vitro. Moreover, profilin depletion/mutations are associated with abnormal phenotypes of microtubules in cells. However, whether profilin regulates microtubules directly, or indirectly through the actin cytoskeleton has remained elusive. Here, Cisterna et al. approached this question by knockout-rescue experiments with different profilin mutants and by analyzing the effects of actin and myosin inhibitors on microtubules. Importantly, they show that in undifferentiated and differentiated (neuron-like) CAD cells, as well as in primary neurons, profilin depletion (or disrupting the actin cytoskeleton by long-term treatment with Latrunculin A) results in increased α -tubulin intensity and acetylation, and provide evidence that these effects of profilin depletion on microtubules occur primarily through the actin cytoskeleton.

The data presented in the manuscript appear of good technical quality, and the study provides important new information on the mechanisms by which profilin controls microtubules in cells. This work also elucidates the mechanisms by which certain profilin mutations contribute to ALS.

Thank you for this positive assessment of our work.

However, there are few points that should be addressed to strengthen the study.

1. The authors state in the Abstract that 'All the observed changes to microtubules were reversible if actomyosin was restored, arguing that PFN1's regulation of microtubules occurs principally through actin'. This is not entirely correct, because this conclusion is only based on blebbistatin washout experiments. Thus, if the authors wish to state that profilin's regulation of microtubules occurs mainly through actin, they should examine if the MT-phenotype of PFY1 knockout cells could be (partially) rescued either by elevating the levels of cytoplasmic actin (see e.g. Suarez et al., *Dev. Cell.* 2015), or by increasing the contractility e.g. by expressing dominant active RhoA in the knockout cells.

This is a fair point. We considered your suggested experiments and have performed a slightly different one that still addresses the concern. The reason why we did not just elevate cytoplasmic actin is because the monomers in PFN1 KO cells do not polymerize well without profilin 1 (Skruber et al. 2020). Previous attempts at expressing more actin did not result in more actin filaments. The dominant active RhoA experiment seemed more feasible, but we were slightly worried about affecting pathways other than non-muscle myosin 2 activation. Therefore, we performed experiments to stimulate actomyosin contractility

directly using the myosin phosphatase inhibitor Caliculin A (Ishihara et al. 1989), which has already been shown to alter microtubule acetylation by Ken Yamada's group (Joo and Yamada 2014). As seen in Figure S3 H and I, applying Caliculin A overnight to PFN1 KO cells at a concentration that causes dramatic changes in F-actin in Control cells (Figure S3 G) has no effect on microtubules.

2. It would be informative to show representative examples of the actin phenotypes of different PFY1 knockout-rescue cells (with different mutant profilins) as an additional supplementary figure. This is important, because apart from the effects of these profilin mutants on F-actin levels, it remained unclear if these mutants have distinct effects on lamellipodial actin filaments, microspikes, or contractile actomyosin bundles.

We have now added super-resolution images of actin in PFN1 KO cells rescued with all of the constructs used in Figure 1D. We have included whole cell images as well as zoomed in regions of the leading edge and internal parts of the cell to demonstrate the effect each construct has on the different actin phenotypes of PFN1 KO cells.

3. The authors should clarify in the legend to Fig. 1D that 'control cells' are PFY1 knockout cells, not wild-type cells (I assume that this was the case).

Thank you for catching this error, the text has been changed.

4. Fig. 2: Links between the actin cytoskeleton and MT-phenotype in the differentiated CAD cells remain unclear, unless the authors will carry out Lat A treatment (or rescue experiments with the profilin R88E mutant) also in the differentiated CAD cells.

We performed the second experiment which you suggested by expressing GFP, GFP-PFN1^{WT}, or GFP-PFN1^{R88E} in PFN1 KO cells and then differentiating them. As shown in Fig. 3 F and G, only PFN1^{WT} was able to rescue the microtubule phenotype.

Reviewer #2

Actin-microtubule crosstalk is an area of emerging interest. The exact molecular mechanisms by which these two cytoskeletal networks exert control over each other, however, still remain largely unresolved. One such mechanism which has recently received attention is the direct binding of ubiquitous actin-binding protein profilin to microtubules.

While profilin has previously been shown to regulate microtubule dynamics in vitro, in vivo implications of this interaction are less clear. The current manuscript shows that in cells, profilin's effects on microtubules are not due to its direct binding to microtubules as has been previously suggested (Henty-Ridilla et al, 2017). The authors here show that profilin-related changes in microtubule dynamics are indirect and mediated by profilin's effects on actin assembly and actomyosin contractility rather than profilin's direct interactions with microtubules. More specifically, they report that these effects of profilin are enhanced in neuronal processes and that they mimic changes in actin cytoskeletal found in neurodegenerative diseases. Authors convincingly show that their results are consistent across multiple

cell lines, differentiated and undifferentiated, and provide insights into how profilin's ALS mutants can affect progenitor cells vs neuronal-like cells.

Data appears to be carefully acquired, rigorously analyzed and presented well. Although the manuscript is well-written overall,

Thank you for this positive assessment of our work.

I think certain parts can be expanded for clarity (see below). While I don't have any major concerns about the manuscript, I mention below a few minor points which I hope will help improve the manuscript.

1. In general, I found both the text and figures to be very dense. I believe this might be a result of the limitations imposed by the report format. As an example, some of the panels in Figure 1 could be moved to supplementary data. Authors could also consider splitting Figure 2 into two separate figures - panels A-H focusing on quantification in knockout vs control, and panels I-M focusing on the effects of perturbations.

We have rearranged Figures 1 and 2 (now Figures 1-3) to help to make them easier to read. Please note that this was done while still conforming to the five figure/three supplementary figure report format and incorporating data from additional experiments requested by Reviewers #1 and #3. However, we feel that the paper does look better now, and the data contained in these figures is easier to access. Thank you for this suggestion.

2. Figure 2D - I am not sure if the image below the graph is helpful since I didn't find any mention of it in the text.

We removed that image and the quantification due to space limitations and because Figure 3 D-F essentially shows the same result.

3. Figure 3 - What was the reason for looking only into the heavy chain subunit? An argument for why authors decided to focus only on the heavy chain would be helpful for non-expert readers.

We have added an explanation of why we looked at neurofilament heavy chain. Briefly, overexpression of this subunit alone causes motor neuron disease in mice (Meier et al. 1999). Additionally, elevated levels of the phosphorylated neurofilament heavy chain are currently one of the most sensitive and specific biomarkers for ALS (Poesen and Van Damme 2018). Delving into the relationship between neurofilaments and other intermediate filaments with PFN1 in ALS models is one of our future goals. Please note that this figure has now been moved to Supplementary Material (Figure S2).

4. Figure 3B- I didn't find any mention of HeLa cells in the main text. I assume they were used as an example of non-neuronal cells. If it is the case, explicitly mentioning this in the text might help.

You are correct and thank you for catching that omission. We have updated the text of the manuscript (Line 167) and the figure legend to describe why HeLa cells were used. Please note that this figure has now been moved to Supplementary Material (Figure S2).

5. Figure 3 - The text in the manuscript describing the figure could be expanded.

We have added additional text to this portion of the manuscript (Lines 258-271).

6. In the results section talking about Figure 4- 'neurites' needs to be deleted from the first sentence.

We have changed this sentence as suggested (Line 177).

7. Figure 4B- Images of α -tubulin and acetyl α -tubulin are mis-aligned i.e. they are not exactly the same fields of view. I assume that the authors have done all the quantification on images that were properly aligned, but still worth a check.

We have changed Figure 4 to have new images (using the images that were published in the first version of the preprint) and made sure that they are properly aligned. This issue did not affect the quantification of the images.

8. Figure 5A - Images of F-actin (top row) and the overlay (lower row) for both control and Blebbistatin treatment are mis-aligned. As mentioned above, they are not the same exact fields of view. Please correct.

We have changed the figure to have new images and made sure that they are properly aligned.

9. I didn't find any mention of supplementary figure 2 in the main text.

We have added text describing it to the manuscript. Please note that this figure is now Fig. S3.

Reviewer #3

Overall, this is a manuscript of interest to the JCB readership.

Thank you.

However, before it can be published, there are a number of points that I would like to raise and which should be considered by the authors to improve their work.

Firstly, as explained in my detailed comments, the paper requires a better argumentation which closely describes existing knowledge of PFN1 and then carefully argues as to whether data presented here provide potential alternative explanations or might even override existing views.

Second, in my view, the heavy reliance on CAD cells weakens the work. What do we know about these CAD cells and their neuritic actin organisation? Do they have the typical actin-spectrin sleeve? Does PFN1 loss affect that sleeve? The clumpy appearance upon PFN1 KO is unexpected and resembles drug-treated axons rather than a system that merely reduces its polymerisation efficiency. As explained in my detailed comments, there are some contradictions regarding axon width when imaged for actin versus microtubules, or when comparing CAD cells to hippocampal neurons. These contradictions should be carefully addressed.

Third, throughout the manuscript, conclusions need to be more precise (see detailed comments). For example, in contrast to what was concluded, LatA-mediated actin removal has been shown in CAD and hippocampal cells to influence tubulin amounts and acetylation, but none of the other parameters has been tested upon LatA application. Conclusions need to be carefully adapted to reflect the actual data, or experiments should be provided to support the current conclusions.

Since the concerns from the last three points were repeated below, we have responded to them individually in that section.

Fourth, it would have been helpful to see some readouts at growth cones where relationships between actin and microtubules are far better understood; also the impact of PFN1 on parameters such as filopodia length or numbers would have helped. Existing images could be re-analysed with this in mind?

Adding analysis of growth cones is a great idea but was very difficult to incorporate into the report-style format of this manuscript, which already contained a lot of data in the allowed figures (there is a 5 main figure/3 supplementary maximum). However, we did add analysis of filopodia in differentiated CAD cell processes to Fig. 3C, which were significantly altered in PFN1 KO cells. Thank you for this suggestion.

DETAILED COMMENTS

p.1, right: Please, explain the conventional function of Profilin before delving into microtubules.

We have now added that profilin is an actin-monomer binding protein that controls filament assembly to the Introduction (Line 54).

p.1, right: A bit more background information on the findings why people believe Pfn1 is a microtubule regulator should be explained, so that the experiments shown here can be held against previous arguments.

We have added more detail about what is known about Pfn1 binding and regulating microtubules to this paragraph (Line 55).

p.1,left: Please, name a few other cross-linker far better studied in neurons: spectraplakins (e.g. <http://doi.org/10.1016/j.semcdb.2017.05.019>) and Drebrin/Eb3 (e.g. <https://doi.org/10.1038/ncb1778>; <https://doi.org/10.1111/jnc.15502>)

We have added the actin-microtubule crosslinkers you have suggested to this statement (Line 44).

p.3, left: I do not understand the sentence 'Similarly, only WT and E117G expressing PFN1 KO cells lost the correlation between the amount of F-actin and microtubules.' given that they restore towards wildtype situation which is at one end of the correlation.

Rereading this sentence, we would agree with you that it was confusing. We have rewritten it for clarity (Line 89).

p.3 left: "while the causative ALS-linked PFN1 mutations have been shown to alter actin assembly in complex ways" - please, explain better. It is not clear what you mean: are different mutants behave different in the same essays, or are the same mutants behaving different in different essays?

You are correct, different mutants have loss and gain of function effects in regard to actin assembly, depending on the specific assay that was used. We have rewritten this statement to provide more detail and enhance clarity (Line 92).

Fig.2 I-M: Please, differentiate into antero- and retrograde transport to see whether dyneins and kinesins are similarly changed in their behaviors.

The way that we did the imaging for this study makes it impossible to tell which direction was anterograde or retrograde, since we only imaged portions of the processes without knowing where the cell body was. We are very interested in this but would like to reserve the details for a future study. We have updated the text of the manuscript to detail that we currently don't know which microtubule motors are predominantly affected. Additionally, there is precedence that both kinesins and dynein should respond similarly to hyperacetylated microtubule bundles (Alper et al. 2014; Balabanian, Berger, and Hendricks 2017; Reed et al. 2006) and we have added this statement and references to the discussion of this result (Line 157).

p.5, right: '~400% increase in number of MTs'; authors measured intensity of tubulin staining and should stick to this readout. They can argue that this likely reflects an increase in numbers and explain the approach they took in Fig.2D; readers should be able to decide as to whether they agree with this calculation. I don't, as also argued later.

We have modified how this result is described as requested (Line 141).

p.5, right: "This highlighted the importance that morphology has in the response to cytoskeletal dysregulation." - I do not understand what the authors want to say. Please, reformulate.

We were trying to convey that the effects on MTs after PFN1 depletion were stronger in neurite-like processes. We have reworded this statement to better reflect that idea (Line 147).

p.5, right, 2nd para: Reformulate the first sentence along the lines of: "The maintenance of neurites requires the antero- and retrograde transport of materials ..."

We have reformatted that sentence as suggested (Line 149).

p.5, left: The fact that neurites become narrower makes me question whether these are really neuron-like cells. For example, has it been shown that they have a periodic actin-spectrin cortex? It is confusing that the microtubules show the opposite trend, i.e. become thicker, as quantified in Fig.2D. Hippocampal neurons shown later on show a thinning of MT bundles. Intensity readings for CAD and hippocampal neurons become therefore difficult to compare and interpret. Something here does not look right.

It has not been shown that CAD cells have a periodic actin-spectrin cytoskeleton. However, this structure does not appear in hippocampal neurons until DIV5 and is not fully formed until DIV8 (Xu, Zhong, and

Zhuang 2013), so even *bona fide* neurons may not have it within the time frame in which we are performing experiments. Further, the thinning of axons in neurons through actomyosin contractility requires the periodic actin-spectrin cytoskeleton (Costa et al. 2020), so experiments performed in our hippocampal neuron cultures would likely not result in neurite thinning. Finally, I think it is fair to be skeptical about how closely the processes of differentiated CAD cells mimic those of actual neurons. This is why we used the last two figures of the manuscript to demonstrate that the main principles learned from CAD cells translated to hippocampal neurons.

Fig.3C,D: explain how measurements were done. Did you measure ROIs or entire cells or neurites? The 'D' label is missing from the Figure.

For undifferentiated cells, the measurements are made on entire cells. For differentiated cells or neurons, the measurements are made using an ROI on the process. The ROI is a standard size and is placed at similar distances from the cell body in each cell. We have added this information to text (Line 141) and provided more detail in the Materials and Methods section. We have also modified the figure as suggested.

p.6, left, conclusion first para: This conclusion cannot be made since the authors used PFD1 KO rather than direct depletion of actin. PFD1 function may directly target MTs in neurites, especially when considering that there are functional differences between undifferentiated and differentiated CAD cells as described by the authors. --- Hippocampal cell experiments could be moved up here to make the point about MTs and acetylation stronger, whereas none of the other parameters (transport, NFs) were ever assessed upon LatA application.

We have modified this statement to reflect that the change in organelle motility was due to loss of PFN1 (Line 163).

p.6, left, 2nd para: Please, explain that these data are not from WB analyses but staining intensities. Explain what exactly was measured (info is also not provided in image), so that the statement 'no process-specific increase' can be understood.

The text has been modified as suggested to make sure the reader knows these were immunocytochemistry experiments (Line 167).

p.6, right, middle. Please, indicate the readouts (i.e. WB and intensity) used to make your statements.

The text has been modified as suggested (Section beginning on Line 191).

p.6, left, end of 2nd para: 'neurofilament accumulations' - this statement is not necessarily correct since the authors do not find pathological aggregates in their cells but proper filaments.

We have removed that phrase.

p.6, right: The last conclusion is vague; I would want to see a clear discussion as to how the results for PFN1 found here, relate to or override previous PFN1 models of direct MT regulation.

We removed the final conclusion from that paragraph and have made the discussion points for each section more clearly reflect our findings. With regards to PFN1 and MT regulation, we state that “In conclusion, while there may be subtle effects on microtubules from PFN1 directly binding tubulin that are missed in our assays, our results (Figs. 1 and 2) provide strong evidence that PFN1 depletion predominantly alters microtubules through an adaptive response caused by the long-term loss of F-actin.” (Line 122).

Fig.4B: The images in different channels are not well aligned, i.e. do not show the exact same position. In contrast to differentiated CA cells, the axons in hippocampal neurons seem to become thinner, i.e. MTs are potentially compacted which might explain the increased staining intensity (rather than an increase in MT number). The image for 500nm LatA application shows the extremely weak tubulin staining on the left, yet acetylation does not reflect this at all. I do not trust that image.

Thank you for catching this, along with Reviewer #2. As described above, all the misaligned images have been replaced. This issue did not affect the quantification of the images.

Please, indicate the identity of the rabbit anti- α -tub antibody.

We have added this information to the Materials and Methods section.

References Cited in Our Response:

- Alper, J. D., F. Decker, B. Agana, and J. Howard. 2014. 'The motility of axonemal dynein is regulated by the tubulin code', *Biophys J*, 107: 2872-80.
- Balabanian, L., C. L. Berger, and A. G. Hendricks. 2017. 'Acetylated Microtubules Are Preferentially Bundled Leading to Enhanced Kinesin-1 Motility', *Biophys J*, 113: 1551-60.
- Costa, A. R., S. C. Sousa, R. Pinto-Costa, J. C. Mateus, C. D. Lopes, A. C. Costa, D. Rosa, D. Machado, L. Pajuelo, X. Wang, F. Q. Zhou, A. J. Pereira, P. Sampaio, B. Y. Rubinstein, I. Mendes Pinto, M. Lampe, P. Aguiar, and M. M. Sousa. 2020. 'The membrane periodic skeleton is an actomyosin network that regulates axonal diameter and conduction', *Elife*, 9.
- Ishihara, H., B. L. Martin, D. L. Brautigam, H. Karaki, H. Ozaki, Y. Kato, N. Fusetani, S. Watabe, K. Hashimoto, D. Uemura, and et al. 1989. 'Calyculin A and okadaic acid: inhibitors of protein phosphatase activity', *Biochem Biophys Res Commun*, 159: 871-7.
- Joo, E. E., and K. M. Yamada. 2014. 'MYPT1 regulates contractility and microtubule acetylation to modulate integrin adhesions and matrix assembly', *Nat Commun*, 5: 3510.
- Meier, J., S. Couillard-Despres, H. Jacomy, C. Gravel, and J. P. Julien. 1999. 'Extra neurofilament NF-L subunits rescue motor neuron disease caused by overexpression of the human NF-H gene in mice', *J Neuropathol Exp Neurol*, 58: 1099-110.
- Poesen, K., and P. Van Damme. 2018. 'Diagnostic and Prognostic Performance of Neurofilaments in ALS', *Front Neurol*, 9: 1167.
- Reed, N. A., D. Cai, T. L. Blasius, G. T. Jih, E. Meyhofer, J. Gaertig, and K. J. Verhey. 2006. 'Microtubule acetylation promotes kinesin-1 binding and transport', *Curr Biol*, 16: 2166-72.
- Skruber, K., P. V. Warp, R. Shklyarov, J. D. Thomas, M. S. Swanson, J. L. Henty-Ridilla, T. A. Read, and E. A. Vitriol. 2020. 'Arp2/3 and Mena/VASP Require Profilin 1 for Actin Network Assembly at the Leading Edge', *Curr Biol*, 30: 2651-64 e5.
- Xu, K., G. Zhong, and X. Zhuang. 2013. 'Actin, spectrin, and associated proteins form a periodic cytoskeletal structure in axons', *Science*, 339: 452-6.

March 12, 2024

RE: JCB Manuscript #202309097R

Dr. Eric Vitriol
Augusta University
Neuroscience and Regenerative Medicine
1462 Laney Walker Blvd.
Augusta 30912

Dear Dr. Vitriol:

Thank you for resubmitting your manuscript entitled "Prolonged depletion of PFN1 or F-actin causes an adaptive response in microtubules." We appreciate your careful and conscientious responses to the multiple specific comments of the reviewers, and upon re-review by one of the expert reviewers, they also felt that you and your co-authors have resolved the concerns. Consequently, we are happy to publish your paper in JCB pending final revisions necessary to meet our formatting guidelines (see details below).

We hope you will agree that the rigorous JCB reviewing process has resulted in superior paper of which you and your co-authors can rightly be quite proud.

A. MANUSCRIPT ORGANIZATION AND FORMATTING:

1) Text limits: Character count for Reports is < 20,000, not including spaces. Count includes the abstract, introduction, results, discussion, and acknowledgments. Count does not include title page, materials and methods, figure legends, references, tables, or supplemental legends. Your manuscript is below this limit but please bear it in mind when revising.

2) Figures limits: Reports may have up to 5 main text figures.

3) Figure formatting: Scale bars must be present on all microscopy images, including inset magnifications. Molecular weight or nucleic acid size markers must be included on all gel electrophoresis. ****Please be sure to add molecular weight markers to the blot in Supplementary Figure 3B.****

4) Statistical analysis: Error bars on graphic representations of numerical data must be clearly described in the figure legend. The number of independent data points (n) represented in a graph must be indicated in the legend. Statistical methods should be explained in full in the materials and methods. For figures presenting pooled data the statistical measure should be defined in the figure legends. Please also be sure to indicate the statistical tests used in each of your experiments (both in the figure legend itself and in a separate methods section) as well as the parameters of the test (for example, if you ran a t-test, please indicate if it was one- or two-sided, etc.).

****Also, since you used parametric tests in your study (e.g. t-tests, ANOVA, etc.), you should have first determined whether the data was normally distributed before selecting that test. In the stats section of the methods, please indicate how you tested for normality. If you did not test for normality, you must state something to the effect that "Data distribution was assumed to be normal but this was not formally tested."****

5) Materials and methods: Should be comprehensive and not simply reference a previous publication for details on how an experiment was performed. Please provide full descriptions (at least in brief) in the text for readers who may not have access to referenced manuscripts. The text should not refer to methods "...as previously described."

6) Please be sure to provide the sequences for all of your primers/oligos and RNAi constructs in the materials and methods. You must also indicate in the methods the source, species, and catalog numbers (where appropriate) for all of your antibodies.

7) Microscope image acquisition: The following information must be provided about the acquisition and processing of images:
a. Make and model of microscope
b. Type, magnification, and numerical aperture of the objective lenses
c. Temperature

- d. imaging medium
- e. Fluorochromes
- f. Camera make and model
- g. Acquisition software
- h. Any software used for image processing subsequent to data acquisition. Please include details and types of operations involved (e.g., type of deconvolution, 3D reconstitutions, surface or volume rendering, gamma adjustments, etc.).

8) References: There is no limit to the number of references cited in a manuscript. References should be cited parenthetically in the text by author and year of publication. Abbreviate the names of journals according to PubMed.

9) Supplemental materials: There are strict limits on the allowable amount of supplemental data. Reports may have up to 3 supplemental figures. At the moment, you meet this limit but please bear it in mind when revising. Please also note that tables, like figures, should be provided as individual, editable files. A summary of all supplemental material (that is, in addition to the supplementary figure legends) should appear at the end of the Materials and methods section.

10) eTOC summary: A ~40-50 word summary that describes the context and significance of the findings for a general readership should be included on the title page. The statement should be written in the present tense and refer to the work in the third person. It should contain "First author name(s) et al..." to match our preferred style.

11) Conflict of interest statement: JCB requires inclusion of a statement in the acknowledgements regarding competing financial interests. If no competing financial interests exist, please include the following statement: "The authors declare no competing financial interests." If competing interests are declared, please follow your statement of these competing interests with the following statement: "The authors declare no further competing financial interests."

12) A separate author contribution section is required following the Acknowledgments in all research manuscripts. All authors should be mentioned and designated by their first and middle initials and full surnames. We encourage use of the CRediT nomenclature (<https://casrai.org/credit/>).

13) ORCID IDs: ORCID IDs are unique identifiers allowing researchers to create a record of their various scholarly contributions in a single place. Please note that ORCID IDs are now *required* for all authors. At resubmission of your final files, please be sure to provide your ORCID ID and those of all co-authors.

14) Please note that JCB now requires authors to submit Source Data used to generate figures containing gels and Western blots with all revised manuscripts. This Source Data consists of fully uncropped and unprocessed images for each gel/blot displayed in the main and supplemental figures. Since your paper includes cropped gel and/or blot images, please be sure to provide one Source Data file for each figure that contains gels and/or blots along with your revised manuscript files. File names for Source Data figures should be alphanumeric without any spaces or special characters (i.e., SourceDataF#, where F# refers to the associated main figure number or SourceDataFS# for those associated with Supplementary figures). The lanes of the gels/blots should be labeled as they are in the associated figure, the place where cropping was applied should be marked (with a box), and molecular weight/size standards should be labeled wherever possible. Source Data files will be made available to reviewers during evaluation of revised manuscripts and, if your paper is eventually published in JCB, the files will be directly linked to specific figures in the published article.

15) Journal of Cell Biology now requires a data availability statement for all research article submissions. These statements will be published in the article directly above the Acknowledgments. The statement should address all data underlying the research presented in the manuscript. Please visit the JCB instructions for authors for guidelines and examples of statements at (<https://rupress.org/jcb/pages/editorial-policies#data-availability-statement>).

B. FINAL FILES:

-- Cover images: If you have any striking images related to this story, we would be happy to consider them for inclusion on the journal cover. Submitted images may also be chosen for highlighting on the journal table of contents or JCB homepage carousel.

Images should be uploaded as TIFF or EPS files and must be at least 300 dpi resolution.

****It is JCB policy that if requested, original data images must be made available to the editors. Failure to provide original images upon request will result in unavoidable delays in publication. Please ensure that you have access to all original data images prior to final submission.****

****The license to publish form must be signed before your manuscript can be sent to production. A link to the electronic license to publish form will be sent to the corresponding author only. Please take a moment to check your funder requirements before choosing the appropriate license.****

Thank you for your attention to these final processing requirements. Please revise and format the manuscript and upload materials within 7-14 days. If you need an extension for whatever reason, please let us know and we can work with you to determine a suitable revision period.

Thank you for this interesting contribution, we look forward to publishing your paper in Journal of Cell Biology.

Sincerely,

Kenneth Yamada, MD, PhD
Senior Editor
Journal of Cell Biology

Tim Spencer, PhD
Executive Editor
Journal of Cell Biology

Reviewer #1 (Comments to the Authors (Required)):

The authors have satisfactorily addressed my previous concerns.